# 20<sup>th</sup> century cooling of the deep ocean contributed to delayed acceleration of Earth's energy imbalance

A. Bagnell [1]✉ & T. DeVries [2,3]

The historical evolution of Earth's energy imbalance can be quantified by changes in the global ocean heat content. However, historical reconstructions of ocean heat content often neglect a large volume of the deep ocean, due to sparse observations of ocean temperatures below 2000 m. Here, we provide a global reconstruction of historical changes in full-depth ocean heat content based on interpolated subsurface temperature data using an auto-regressive artificial neural network, providing estimates of total ocean warming for the period 1946-2019. We find that cooling of the deep ocean and a small heat gain in the upper ocean led to no robust trend in global ocean heat content from 1960-1990, implying a roughly balanced Earth energy budget within $-0.16$ to $0.06$ W m$^{-2}$ over most of the latter half of the 20th century. However, the past three decades have seen a rapid acceleration in ocean warming, with the entire ocean warming from top to bottom at a rate of $0.63 \pm 0.13$ W m$^{-2}$. These results suggest a delayed onset of a positive Earth energy imbalance relative to previous estimates, although large uncertainties remain.

[1] Interdepartmental Graduate Program in Marine Science, University of California, Santa Barbara, USA. [2] Department of Geography, University of California, Santa Barbara, USA. [3] Earth Research Institute, University of California, Santa Barbara, USA. ✉email: abagnell@ucsb.edu

Global climate change is driven by imbalances in Earth's energy budget due to both anthropogenic and natural influences[1,2]. Estimating historical changes in Earth's energy imbalance (EEI) is essential for accurately quantifying climate sensitivity to greenhouse gas emissions, benchmarking climate models used in making future climate projections, and for understanding the contribution of natural events and climate patterns to modulating the global climate response to anthropogenic forcing[2,3]. The ocean is currently the largest energy reservoir in the Earth's climate system and is responsible for absorbing and storing more than 90% of the excess heat in the Earth system that results from anthropogenic climate change[2–4]. Thus, measurements of the global ocean heat content (OHC) over time provide one of the best ways of estimating historical trends in the EEI[2–4].

Historical changes in global OHC can best be reconstructed from in situ temperature observations. Over the past 15 years, the Argo program[5] has deployed thousands of autonomous floats which provide continuous observations of the temperature in the upper half of the ocean, to a depth of 2000 m. This has allowed for a convergence in estimates of OHC over the last fifteen years[2,5] and increased confidence in calculations of the ongoing EEI in light of independent confirmation from modern satellite observations[2,6,7]. However, several challenges exist for reducing uncertainty in estimates of total ocean warming and extending it over longer time periods. First, the deep ocean below 2000 m remains poorly observed, even during the Argo era, which leads to additional uncertainty on current estimates of total warming. While absolute temperature changes in the deep ocean are small[8], the large volume of the ocean below 2000 m makes it a potentially meaningful contributor to the global heat inventory. Repeat hydrographic sampling indicates that the deep ocean may be warming significantly in some regions[9], particularly the Southern Ocean[8], whereas other regions may still be cooling as a response to cold periods in the past millennium[10], making it critical to include the heat content of the deep ocean in global estimates of ocean warming. The second issue is that, prior to 2005, data collection was conducted primarily by scientific research vessels and ships of opportunity, leaving areas outside of major trade routes or research transects with few direct observations[2,11]. This leaves large gaps in the observational record that must be filled in order to estimate OHC.

Several methods have been devised to overcome these gaps in ocean temperature observations and to produce estimates of historical changes in OHC. One common approach applies objective mapping to interpolate the sparse temperature records in space and time[11–13]. However, while these objective mapping products can reconstruct ocean temperatures back to ~1950, they do not extend below 2000 m due to the sparse sampling at these depths. Dynamical data-assimilation models offer an alternative approach to objective mapping and provide full-depth estimates of OHC[14,15], but data sparsity means these models are poorly constrained at depth, leading to large cross-model variance[15]. Another approach based on the passive transport of surface temperature anomalies into the interior ocean[10,16] can also reconstruct full-depth temperature anomalies and OHC changes, but relies on the potentially incorrect assumption of steady-state circulation[16] and is sensitive to the initial condition used in the simulation[10,16] and to poorly known surface ocean temperatures dating back several millennia[10]. Finally, statistical methods have been used to detect large-scale trends in the deep ocean temperature from repeat hydrographic sampling[9], but these have coarse spatial resolution and do not cover the period prior to the mid-1980s. An interpolation product based on in situ temperature data that covers the deep ocean below 2000 m, allowing for a full-depth OHC estimate, remains crucial to reliably estimating historical changes in EEI[7,17].

Here, we interpolate historical ocean temperature data using an autoregressive artificial neural network (ARANN) to produce a single consistent estimate of the top-to-bottom OHC change for 1946–2019 using in situ temperature data from the World Ocean Database[18]. This approach (Supplementary Figs. 1–2) adapts an established machine learning method to perform an iterative autoregression that adjusts spatio-temporal correlation scales over time from the in situ temperature data itself, and effectively propagates information from well-sampled times and regions to more sparsely sampled areas to produce global maps of temperature anomalies at roughly annual resolution (Supplementary Fig. 3). This approach is robust to sparse data, allowing our estimates of OHC change to be extended below 2000 m to the seafloor (Fig. 1). We have tested the method on datasets from two ocean models used in the Climate Model Intercomparison Project Phase 6 (CMIP6)[19,20], demonstrating the ability to accurately reconstruct OHC changes on both global and basin scales (Supplementary Figs. 4–7) at all depths of the ocean, and to recreate modeled temperature anomalies at spatial scales of ~1000 km or larger (Supplementary Figs. 8–11), even in the presence of realistic geophysical noise that is present in the observations but not the models (Supplementary Figs. 4–12). We apply the ARANN in an ensemble approach designed to take into account sources of uncertainty arising from the sparse distribution of temperature observations[2,11], documented instrument biases[21–24] (Supplementary Fig. 13), and choice of reference climatology used to define the temperature anomalies[25,26] (Supplementary Fig. 14).

Four instrumental bias corrections and six decadal climatologies are combined with random selections of temperature data to produce the 240 ensemble members used in this study. This ensemble is used to assess the uncertainty of our OHC reconstruction and provide bounds on our estimates of ocean warming. All estimated warming rates come from fitting a linear trend to the mean ARANN OHC estimate and uncertainties in these rates are calculated by taking 2 standard deviations across all ensemble members. Where ranges are given, these compare the mean ARANN estimate to other products. For simplicity, OHC estimates from other studies are not plotted with their respective confidence intervals, as the methodology for calculating these varies by study, but they generally possess uncertainty levels similar to those provided by the ARANN.

## Results

**Global and basin-scale OHC changes.** Estimates of the global full-depth OHC from the ARANN method show that there was no net ocean warming during the four decades from 1950 to 1990, but instead the OHC fluctuated by ~50 ZJ on roughly decadal timescales (Fig. 1a). However, since 1990 there has been a rapid acceleration in ocean warming, with the ocean gaining 303 ± 56 ZJ of thermal energy in the past three decades (Fig. 1a). This temporal pattern is roughly consistent throughout the water column, with minor warming prior to 1990 in the upper 700 m, no warming in the 700–2000 m depth range, and cooling in the deep and abyssal layers below 2000 m (Fig. 1b–d). Warming rates accelerated substantially after 1990 throughout the entire water column, with the deep ocean switching from cooling to warming after 1990 (Fig. 1b–d).

Passive transport methods[10,16] that propagate surface temperature anomalies to the deep ocean using steady-state ocean circulation patterns provide internally consistent estimates of full-depth OHC that can be directly compared to ARANN, after adjusting their baselines to coincide with the ARANN estimate during the Argo era (2005 onwards) (Fig. 1a). These passive transport estimates differ from the ARANN and from each other. Both show an earlier onset of ocean warming than the ARANN,

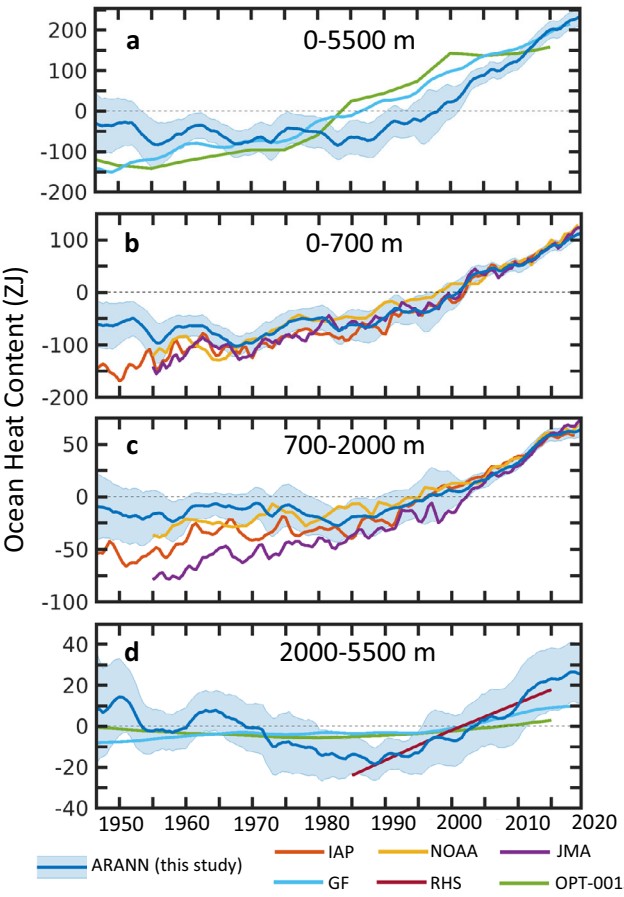

**Fig. 1 Estimates of global ocean heat content.** Estimates of ocean heat content (OHC) changes for **a** the global ocean from the surface to seafloor, **b** the upper 700 m of the ocean, **c** the depth range 700–2000 m, and (**d**) the depth range 2000–5500 m. The mean estimates derived from this study (ARANN, blue) are shown with shading covering two standard deviations from the mean over the 240 ARANN ensemble members. The zero anomaly is defined such that the mean OHC of the ARANN estimate for the period 1946–2019 is zero. Also shown are the mean OHC anomaly from the IAP[11] (red), NOAA[12] (yellow), and JMA[13] (purple) objective mapping products, which cover the 0–2000 m depth interval as shown in (**b**)–(**c**). These products have been adjusted to the mean ARANN OHC anomaly for 2005–2019. Shown for **a** the full ocean depth and **d** the deep ocean are OHC anomalies from passive ocean heat uptake models using Green's functions (GF)[16] (light blue) and an optimized mixing model (OPT-0015)[10] (green). The passive ocean heat uptake products are adjusted to the mean ARANN anomaly for 1955–1985. Repeat hydrographic sampling (RHS) gives temperature trends since 1985 in the deep ocean[9] (maroon; **d**). The RHS method gives a linear trend from 1985 to 2000 and from 2000 to 2015, which has been adjusted to the mean ARANN anomaly for 1985–2015.

with full-depth ocean warming starting in the mid-1970s for the optimized mixing model (OPT-0015)[10] and going back to the 1950s in the Green's function (GF)[16] product (Fig. 1a). The OPT-0015 method shows very little ocean warming prior to the mid-1970s, in agreement with the ARANN reconstruction, while the GF method suggests a nearly constant ocean warming trend throughout the past ~70 years.

Estimates of OHC from objective mapping products can be compared to the ARANN estimates in the upper 2000 m (Fig. 1b–c). There is broad agreement about the total change in OHC among the objective mapping products and with the

ARANN ensemble for much of 1980–2019, with the JMA[13] estimate on the very edge of the ARANN uncertainty range for the 700–2000 m depth interval (Fig. 1c). Prior to 1980, the mapping methods diverge somewhat in their predictions over the upper 2000 m, and the disagreement is most pronounced for the earliest time periods. After adjusting the OHC anomalies of the objective mapping estimates to the mean ARANN 0–2000 m OHC value over 2005–2019, the ARANN OHC in 1955 is 71 ± 58 ZJ greater than that estimated by the IAP[11] product, 51 ± 58 ZJ greater than the NOAA[12] product, and 114 ± 58 ZJ greater than the JMA[13] product (Fig. 1b–c). The large spread among OHC products prior to 1980 is primarily due to increased data sparsity in this period, but the choice of reference climatology also plays a role in enhancing uncertainties in the ARANN during this time period. For years prior to 1970, mean ARANN OHC over the 0–2000 m depth interval can vary by as much as 67 ZJ when using different reference climatologies (Supplementary Fig. 14), which is a source of uncertainty that has generally been neglected in the other mapping products. Despite these uncertainties, all mapping methods show an acceleration in OHC uptake over time in the 0–2000 m interval. For the mapping products, the underlying subsurface temperature data creates strong constraints that reduce the variance across methods over the last several decades of OHC estimates (Fig. 1b–c), in contrast to the passive transport products where differences in methodology have large impacts on the inferred OHC trends (Fig. 1a).

The ARANN yields a global interpolation of deep subsurface ocean temperature data and shows a cooling trend from 1950 to 1990, representing a reduction of OHC by 26 ± 16 ZJ (Fig. 1d), mainly canceling out the small heat gain in the upper 700 m and contributing to the negligible warming of the global ocean estimated by the ARANN during this time period (Fig. 1a). The passive transport methods both predict almost negligible changes in deep OHC during this period, with the GF method showing slight warming and OPT-0015 showing slight cooling (Fig. 1d). The ARANN predicts that the deep ocean has warmed significantly since 1990, gaining 48 ± 19 ZJ, at a rate that closely matches the estimates from repeat hydrographic surveys[9] (RHS). Over the past 30 years the deep ocean has gained back all of the heat lost since 1950, arriving at possibly its warmest level over the entire 75-year record (Fig. 1d). This rapid warming of the deep ocean is contrary to the slow rise in OHC implied by the passive transport models that use steady ocean circulation[10,16], suggesting that transient features of the deep ocean circulation are important for contributing to the warming over the recent three decades[8,27].

Clear regional differences in ocean warming rates emerge on ocean basin scales (Fig. 2). In the upper 700 m, the Atlantic Ocean has warmed the most of all the major ocean basins, showing sustained warming since the 1950s and accounting for more than one third of the total warming in the upper 700 m. Objective mapping methods and the ARANN agree for the period after 1980 in the shallow Atlantic, however, the ARANN produces 7–10 ZJ less warming than the objective mappings prior to 1985 (Fig. 2a). The sustained warming of the Atlantic Ocean has penetrated into the intermediate layers (700–2000 m), but methods disagree on the amount of warming prior to 2005, with the JMA and IAP estimates putting as much as 10 ZJ more warming into the intermediate Atlantic than the ARANN and NOAA methods for much of the record (Fig. 2b). The ARANN reconstruction of deep Atlantic OHC (below 2000 m) shows a slight cooling trend until ~2005 then subsequent warming. This reversal from cooling to warming trends is also captured by the repeat hydrography data (Fig. 2c). However, the passive transport method OPT-0015 shows an accelerated warming of the deep

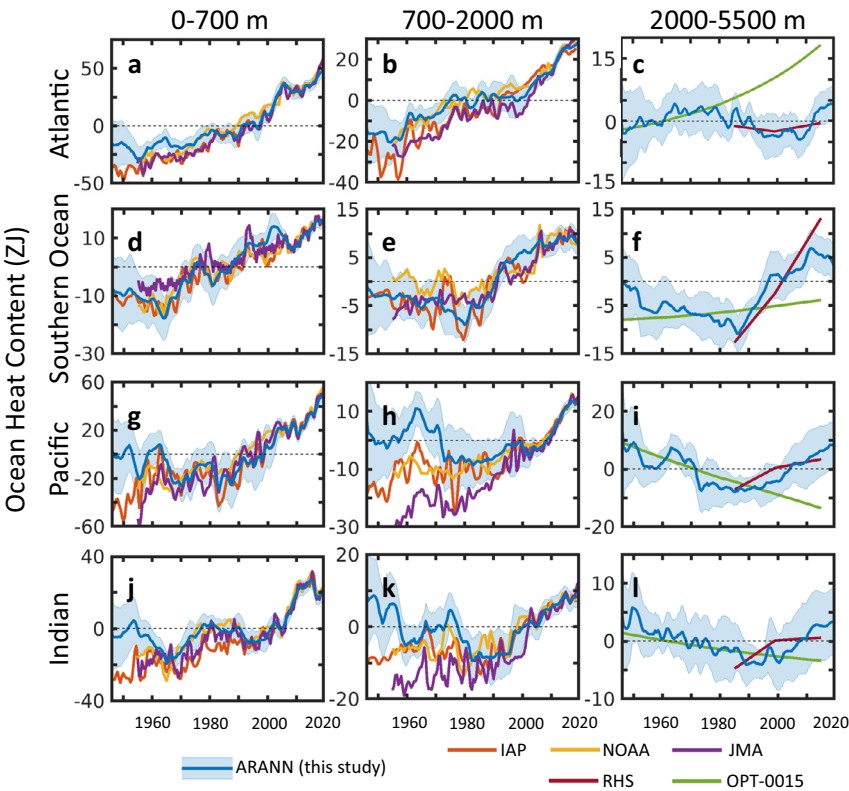

**Fig. 2 Ocean heat content for the major ocean basins.** Basin-scale ocean heat content (OHC) anomalies for the Atlantic Ocean at depth intervals, **a** 0–700 m, **b** 700–2000 m, **c** 2000–5500 m; for the Southern Ocean at depth intervals, **d** 0–700 m, **e** 700–2000 m, **f** 2000–5500 m; for the Pacific Ocean at depth intervals, **g** 0–700 m, **h** 700–2000 m, **i** 2000–5500 m; and for the Indian Ocean at depth intervals, **j** 0–700 m, **k** 700–2000 m, **l** 2000–5500 m. Ocean basins are defined using the World Ocean Atlas mask[63], with the Southern Ocean considered everything south of 50° S. OHC anomalies and uncertainties are computed as in Fig. 1 and compared with previous reconstructions as in Fig. 1. The Green's Function (GF) method does not provide basin-scale estimates of OHC and is omitted in the comparison here.

Atlantic over the entire time period (Fig. 2c), which may result from biases in its estimated ventilation rates. Passive transport of recent surface warming into the internal Atlantic via a steady-state ocean circulation overlooks natural variability in rates of overturning and deep water formation[28–30], possibly explaining an overestimate of the warming trend in the OPT-0015 versus those of subsurface observations such as ARANN and RHS.

Fingerprints of ocean circulation changes are also apparent in the spatial distribution of warming rates (Fig. 3). Prior to 1990, the upper 700 m of the subpolar and polar North Atlantic was cooling, while the Gulf Stream extension region was warming (Fig. 3a), consistent with trends that have previously been identified as fingerprints of a slowdown in the Atlantic Meridional Overturning Circulation (AMOC) and reduced deep convection in high-latitude deep water formation regions[32,33]. Since 1990 there has been coherent strong warming throughout most of the Atlantic basin, except for a small patch of cooling in the center of the North Atlantic subpolar gyre (Fig. 3d). Warming of the North Atlantic from 1990 to 2005 has previously been linked to a surge in the AMOC after 1990[31], although this has been followed by a decline in the AMOC after 2005 and cooling of the subpolar gyre[33], potentially contributing to the cooling trend identified over this longer period in the central subpolar gyre (Fig. 3d). After 1990 there is also pronounced warming at mid depths (700–2000 m) throughout most of the Atlantic, concentrated more strongly in the subpolar north and south Atlantic (Fig. 3e). This is consistent with the mean overturning circulation transporting surface warming to intermediate waters, since these regions are close to the formation regions for North

Atlantic Deep Water in the North Atlantic and Antarctic Intermediate Waters in the South Atlantic[34].

Like the Atlantic, the Southern Ocean (defined here as south of 50° S) has been warming consistently since 1960 in the upper 700 m, a trend seen across multiple reconstruction methods (Fig. 2d). In the intermediate layers (700–2000 m), large differences in sub-decadal variability across these methods reveal the impact of sparse temperature sampling in the region, but the consensus across methods is that warming of Southern Ocean intermediate waters started in the 1980s, with little warming before that (Fig. 2e). The ARANN reconstruction shows a cooling trend in the deep Southern Ocean (>2000 m) until ~1985, followed by rapid warming thereafter (Fig. 2f). The post-1985 warming trends in the deep Southern Ocean in the ARANN generally agree with the trends derived from repeat hydrography, whereas the OPT-0015 passive transport method shows a very small warming trend over the entire 1946–2019 period (Fig. 2f). The spatial distribution of warming in the deep Southern Ocean shows that the rapid warming over the past three decades is concentrated along the Antarctic margin, where Antarctic Bottom Waters form in the Weddell Sea, Ross Sea, and other marginal seas along the Antarctic shelf[35,36] (Fig. 3f).

The Southern Ocean is a key global heat sink over the past three decades. Most regions of the Southern Ocean have been warming consistently since 1990 (Fig. 3d–f), and the 0–700 m, 700–2000 m, and 2000–5500 m depth intervals have all experienced roughly the same amount of OHC change during this period (Fig. 2d–f). Currently, the entire Southern Ocean from surface to seafloor sits at its warmest levels since at least the 1950s. This rapid warming of the Southern Ocean has been accompanied by a general asymmetrical

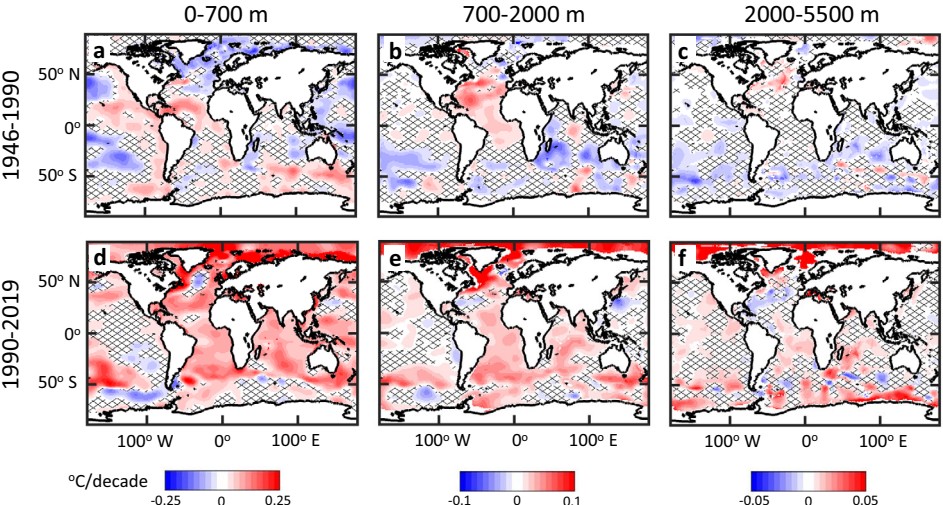

**Fig. 3 Linear warming rates.** Spatial maps of linear warming rates for the period 1946–1990 at depths: **a** 0–700 m, **b** 700–2000 m, and **c** 2000–5500 m, and for the period 1990–2019 at depths: **d** 0–700 m, **e** 700–2000 m, and **f** 2000–5500 m. Warming rates were estimated by averaging temperature anomalies over each depth interval, then applying a linear least-squares fit to the temporal trend of temperature at each 1° grid cell for the specified time periods. Areas without significant trends (95% confidence interval) are cross-hatched. Uncertainties were estimated by calculating the sum of the error of the linear fit and the cross-ensemble uncertainty of the warming rates at each grid cell for the various ARANN ensemble members.

warming over the past decade that has favored heating of the Southern Hemisphere. For the period 2005–2019, the ARANN estimates that the Southern Hemisphere accounted for 68% of global OHC change in the top 700 m, 54% for 700–2000 m, and 81% below 2000 m. This asymmetrical warming is consistent with previous analyses that linked the redistribution of heat to internal climate variability[37]. This appears to be a recent change in the pattern of ocean heat uptake, however, as Northern Hemisphere waters have consistently warmed over the entire 75-year record of the ARANN while Southern Hemisphere waters cooled on average prior to 1990 (Fig. 3).

Warming of the Atlantic and Southern Oceans above 2000 m prior to 1990 was counterbalanced by slower rates of warming or by cooling of the Indian and Pacific Oceans over this time period (Fig. 2g–h, j–k). The Indian and Pacific Oceans show little trend in heat content in the upper 700 m from 1950 to 1990 for the ARANN, while other mapping products show very slight warming over this period. Instead, this period is mostly marked by decadal-scale oscillations in OHC in the ARANN reconstruction, which also appears to some extent in the objective mapping products (Fig. 2g, j). Identifying the mechanisms responsible for these oscillations is beyond the scope of this study, but these could be related to changes in upper-ocean overturning circulation associated with the Interdecadal Pacific Oscillation (IPO)[38], which has been ascribed to changes in the strength of Pacific trade winds that affect eastern equatorial upwelling[39], as well as modifications to the winds in the tropical North Pacific[40]. These factors also appear to modulate heat transport into the Indian Ocean via the Indonesian Throughflow[40,41]. Below 700 m, the OHC estimates of the various mapping products greatly diverge prior to 1990. ARANN indicates that the mid-depth Pacific and Indian Oceans were cooling up until 1990, whereas the IAP and NOAA products show little trend over this period and the JMA method produces unabated warming throughout the record, leading to large disagreements across methods totaling ~45 ZJ of difference in the warming summed over these two basins (Fig. 2h, k).

The ARANN reconstruction also shows that the deep and abyssal layers of the Pacific and the Indian Ocean were cooling up until ~2000 (Fig. 2i, l). This cooling trend agrees well with the OPT-0015 multi-millennial passive ocean heat uptake reconstruction, which produces a 20th-century cooling trend in the deep Pacific and Indian Oceans in response to cold surface temperatures during the Little Ice Age that occurred from the 14th-19th centuries[10]. This cooling trend has been preserved into the 20th century due to the deep ocean's long overturning timescales[10].

Over the past two decades, the Pacific and Indian Oceans have warmed substantially throughout the water column, contributing ~52% of the global OHC change since the year 2000. This warming has been concentrated in the upper 700 m, in agreement with objective mapping reconstructions (Fig. 2g, j). While this recent warming is coherent across most of the Indian Ocean, the warming is more concentrated in the central and western Pacific Ocean, with a slight cooling trend in the eastern Pacific (Fig. 3d). This Pacific pattern is consistent with findings of enhanced meridional temperature gradients in the tropical Pacific Ocean[42] which could help explain a shift toward strong basin-wide El Nino events during recent decades[43]. The ARANN reconstruction also shows substantial warming since 2000 of the mid-depth (700–2000 m) as well as the deep ocean (below 2000 m), in general agreement with objective mapping approaches in the mid-depth layers and with trends derived from RHS in the deep ocean (Fig. 2 h–i, k–l). However, the recent warming of the deep Pacific and Indian Oceans found by the ARANN is contrary to the continued slow cooling implied by passive heat uptake in the OPT-0015, suggesting that the deep ocean warming since 2000 is related to changes in the transport of deep and bottom waters[27,44].

**A shift in EEI.** The time derivative of the full-depth OHC, dOHC/dt, should very closely track the EEI, since all other heat sinks in the climate system are currently an order of magnitude smaller than the ocean[2,3]. Periods of positive dOHC/dt represent ocean warming and a positive net energy imbalance at the top of the atmosphere, while negative dOHC/dt represents ocean cooling and a negative EEI (Fig. 4). The EEI itself reflects a combination of positive anthropogenic forcing due to greenhouse gas emissions and negative forcing due to anthropogenic aerosols[1], as

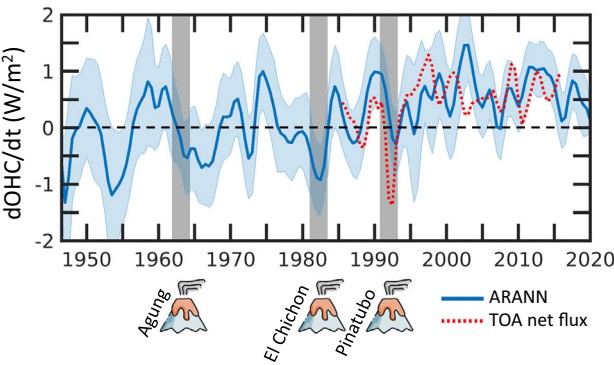

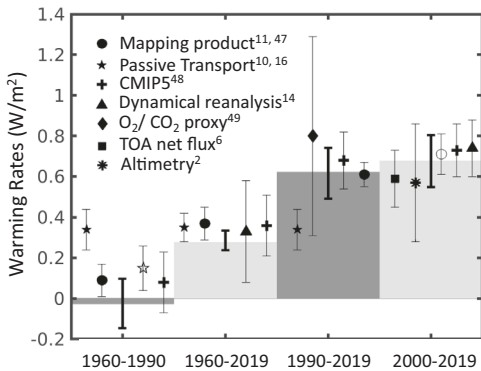

**Fig. 4 Rate of change in global ocean heat content.** Time derivative of the global full-depth ocean heat content (dOHC/dt) from 1946-2019 (blue; mean) with uncertainty (shading; 2 standard deviations) across 240 ARANN ensemble members. The dOHC/dt is computed using a centered difference in time of the global OHC and applying a 1-2-1 filter to smooth the result. This warming rate is divided by the surface area of the Earth so that it can be interpreted as the ocean component of the Earth Energy Imbalance (EEI). This is compared to the top of atmosphere (TOA) net radiative flux[6] (red dashed) for years 1985-2016. The timing of three major volcanic eruptions is represented by icons with the corresponding volcano's name. Vertical gray bars cover 12 months prior to and 18 months after each eruption to account for the imperfect time resolution of the ARANN dOHC/dt (~12 months) and the e-folding time of volcanic aerosols in the stratosphere[64].

**Fig. 5 Global warming rates derived from various methods.** Linear ocean warming rates derived from this study (vertical gray bars) with 95% confidence intervals (bold error bars) computed by taking the minimum and maximum warming rates of the middle 95% of 240 ARANN ensemble members. These are compared to other published estimates and uncertainties (symbols and error bars) derived from mapping methods (filled circle[11], open circle[46]), passive transport products (filled star[16], open star[10]), dynamical reanalyses[14] (triangle), satellite altimetry[2] (asterisk), CMIP5 climate models[47] (cross), top of atmosphere (TOA) net radiative flux[6] (square), and the chemical composition of the atmosphere[48] (diamond). Each symbol is associated with one of the gray bars and covers approximately the same time interval.

well as natural forcing by volcanic eruptions[1,3] and natural variability due to internal dynamics of the climate system[3,4]. We find that over the entire 1946–2019 period covered by our OHC reconstruction, there are more than a dozen transitions between positive and negative values of the dOHC/dt (Fig. 4). Some of the most negative dOHC/dt values are coincident with major volcanic eruptions over the past 50 years, including Agun, El Chichon, and Pinatubo[3]. However, there are more oscillations in the dOHC/dt than can be associated with volcanic aerosol forcing, and in the case of El Chichon the timing of the eruption occurs at a local minimum in the dOHC/dt instead of just prior, indicating that volcanic aerosol forcing may not always dominate over natural variability in the EEI. This supports prior studies that point to internal modes of climate variability such as ENSO[2,7] and the IPO[4,38], or natural variability in solar irradiance[45] as factors influencing sub-decadal changes in the EEI for this 75-year record.

Prior to 1990 the dOHC/dt oscillates around zero, averaging $-0.04 \pm 0.11 \, \mathrm{W \, m^{-2}}$ for the period 1946–1990. Without taking into account the deep ocean cooling during this period, the average dOHC/dt would be $0.01 \pm 0.09 \, \mathrm{W \, m^{-2}}$. An upward shift in the dOHC/dt is noticeable in the mid-1990s (Fig. 4), after which the average warming rate in the ARANN OHC reconstruction is nearly always positive, averaging $0.67 \pm 0.13 \, \mathrm{W \, m^{-2}}$ for the period 2000–2019. The timing of this shift, the magnitude of the implied EEI, and the temporal variability of the EEI agree well with estimates of the top of atmosphere (TOA) net radiative flux[6] for the period 1985–2016. The only time periods where the TOA net radiative flux lies outside 2 standard deviations of the ARANN-estimated EEI are in the early 1990s just after the Pinatubo eruption, when the TOA radiative flux is more negative than the dOHC/dt, and during the early 2000s when the dOHC/dt shows an upward jump that is opposed to a drop in the TOA net radiative flux (Fig. 4).

The magnitude of the ARANN-estimated dOHC/dt from 2000 to 2019 agrees within 2 standard deviations with numerous other estimates for this period, including those based on dOHC/dt from objective mapping products[11,46], ocean reanalysis products[14], and CMIP5 hindcast models[47], as well as estimates of EEI from satellite altimeter and gravity data[2] and top-of-atmosphere radiative fluxes[6] (Fig. 5). Prior to the year 2000, warming rates estimated from our full-depth OHC reconstruction mostly agree within their respective uncertainties with previous estimates from the IAP mapping product[11], passive transport methods[10,16], CMIP5 hindcast models[47], and an independent estimate of EEI for the period 1990–2016 from measurements of atmospheric composition[48]. For the period 1960–1990, the ARANN approach estimates essentially no warming ($-0.03 \pm 0.12 \, \mathrm{W \, m^{-2}}$), implying a roughly balanced Earth energy budget over this time period. Objective mapping, passive transport methods, and climate models have generally estimated small warming rates over the 1960–1990 period, but mostly overlap with the ARANN estimates within their respective uncertainties (Fig. 5). Almost all methods of estimating EEI agree on an acceleration of the EEI in recent decades, particularly since 1990 (Fig. 5). A notable exception is the GF passive transport method, which maintains a steady warming rate across the entire 1960–2019 period, implying an important role for ocean circulation changes in controlling the acceleration of the EEI over recent decades. Overall, the ARANN results support a broad consensus across almost all products of accelerated warming over time (Fig. 5), but they also suggest that previous estimates of ocean warming may have been biased too high prior to 1990, in part due to the neglect of deep ocean cooling. Including the effects of deep ocean cooling, as determined by the mean estimate of the ARANN, would lower the rates of ocean warming prior to 1990 determined by previous objective mapping approaches[11–13] by 18–32%.

## Discussion

The ARANN reconstruction of full-depth OHC provides an internally consistent framework for monitoring EEI over time, showing that the Earth energy budget was in quasi-equilibrium, with substantial decadal variability, for the four decades from 1950 to 1990. The warming rate from the ARANN does not differ

from that derived by objective mapping methods with statistical significance, and previous studies already support a slower ocean warming rate for the 1950–1990 period relative to the 21st century (Fig. 5). However, due to the combination of a smaller estimated change in 0–2000 m OHC for 1950–1990 and the contribution of deep ocean cooling, the ARANN implies a stronger and later shift toward accelerated EEI than previously recognized, and raises the question as to what may have caused this climate shift.

Anthropogenic radiative forcing has remained positive and continued to grow in magnitude over the past century[1], so the lack of global ocean warming implied by the ARANN results over the period from 1950 to 1990 may seem counterintuitive at first. However, Earth's climate system is not currently at equilibrium. Due to the timescales of overturning in the ocean, propagating the entire forced climate signal from the surface to the interior may require decades to centuries to manifest as signals in the deep OHC[8,10], implying that the EEI is modulated by changes in external forcing on multi-decadal time-scales. In the deep ocean, cooling of the Pacific and Indian over much of the 20th century could result from a past climate event such as the Little Ice Age[10]. A cooling trend that derives itself from long-term modes of climate variability[49] would not be reflected in any of the components of the external radiative forcing budget for the 20th century.

Nonetheless, deep ocean cooling does not entirely account for the near zero warming trend in OHC prior to 1990, especially when considering that the 0–2000 m interval shows minimal change in the ARANN OHC estimate as well, averaging just $0.03 \pm 0.09$ W m$^{-2}$ from 1960 to 1990. The difference between the ARANN and the IAP reconstruction[11] of OHC in the upper 2000 m, is similar in magnitude to the ARANN estimate of deep ocean cooling (Supplementary Fig. 15), and in general the spread across OHC estimates in the top 2000 m is larger than the deep ocean cooling trend estimated by the ARANN (Fig. 1b–c). This spread indicates large uncertainties related to methodological differences in estimating OHC over the latter half of the 20th century. However, if the ARANN estimate of minimal upper ocean warming prior to 1990 is correct, it could indicate that anthropogenic or volcanic aerosol effects are larger than currently estimated for this time period[50] or that the transient climate response to anthropogenic forcing is affected by regional feedbacks arising from the pattern of ocean heat uptake[51,52]. Changes in the ocean overturning can also affect the EEI by modifying the rate of ocean heat uptake[53], which could also lead to discrepancies between radiative forcing and upper ocean warming.

The recent accelerated warming since 1990 implied by the ARANN is consistent with the dominant effects of anthropogenic greenhouse gas forcing and negligible volcanic aerosol forcing[1,54], as well as estimates of increased radiative forcing[55] during the past three decades. Due to improved ocean temperature sampling over the past several decades, there is high confidence that the top 2000 m of the ocean have been gaining heat at an accelerating rate, as indicated by the convergence of OHC estimates across methodologies during this time period (Fig. 1b–c). In addition, the ARANN results suggest that the deep ocean below 2000 m has added $48 \pm 19$ ZJ since 1990, or about 10–28% of the ocean warming above 2000 m during this period, significantly contributing to the accelerating EEI in recent decades. This contribution is larger than that from non-ocean components of the Earth energy budget, including the land surface, cryosphere, and atmosphere, which together account for ~$27 \pm 8$ ZJ of warming since 1990[56].

In all, the results presented here show that deep ocean cooling during the latter half of the 20th century has given way to deep ocean warming over the past three decades, contributing to a delayed response of the EEI to contemporary radiative forcing

effects. If this recent shift toward warming of the deep ocean continues, it will have implications for Earth's climate for decades to centuries to come due to the long overturning timescales of the deep ocean. Continued monitoring of the global OHC, and improved resolution of deep ocean temperature changes, will be key for developing accurate forecasts of Earth's energy budget and future climate change.

## Methods

**Observational datasets and data processing.** We used a dataset of historical ocean temperature observations from 1946 to 2019 consisting of individual temperature casts from the World Ocean Database (WOD) 2018[18] (Supplementary Fig. 1a, Step 1). We used temperature casts from multiple instruments, including mechanical bathythermographs and expendable bathythermographs (MBT and XBT), ocean station data (OSD), conductivity-temperature-depth (CTD) profiles, autonomous profiling floats (PFL), and autonomous pinniped bathythermographs (APB). We quality controlled these data using the strictest quality control procedures of the World Ocean Database (WOD), which exclude any data with a flag other than 0. In addition, we excluded casts that have <5 discrete temperature samples, while also requiring that one of these samples occurs in the top 100 m. These extra steps reduce the possibility of including data from casts that have had much of their data removed by flags, casts that had insufficient samples to be properly quality controlled to begin with, or casts where much of the data was removed in the near-surface due to an excessive vertical temperature gradient (0.7 °C m$^{-1}$ for the WOD).

Next, individual casts were linearly interpolated to the 102 standard depths of the World Ocean Atlas[63] (WOA) grid. Before the data from the various instrument types were combined, systematic errors in the bathythermographs were corrected (Supplementary Fig. 1a, Step 2) using some of the best available calibration methods (see Supplementary Fig. 13). Then, these data were binned to the WOA grid at monthly resolution based on the year and calendar month of their collection (Supplementary Fig. 1a, Step 3). We binned using the median of observations within each grid cell, to reduce the influence of outliers.

After binning the temperature data to the WOA grid, we subtracted a monthly temperature climatology to create a field of monthly temperature anomalies (Supplementary Fig. 1a, Step 4). For this step, we used one of six WOA decadal climatologies covering years [1955–64, 1965–74, 1985–94, 1995–2004, 2005–2017]. These climatologies are monthly in the top 1500 m and seasonal below that. The choice of climatology produces a difference in the mean total OHC change from 1946 to 2019 of up to 49 ZJ. The impact of climatological choice is small for the final three decades of the OHC record but has a more significant impact on the global OHC estimate in the deep ocean and further back in time (see Supplementary Fig. 14).

Finally, we smoothed the resulting monthly anomaly maps using a 12-month moving average, and then binned the smoothed monthly anomaly maps to 6-month time intervals that span either Jan–Jun and Jul–Dec (Supplementary Fig. 1b), or Apr–Sep and Oct–Mar. The choice of either Winter/Summer-centered or Spring/Fall-centered anomalies also enters into our ensemble, but has little impact on the final results. The end result is 148 three-dimensional temperature anomaly fields spanning 1946–2019 at 6-month time intervals. Each anomaly field (except for the first and last) contains temperature data from within an 18-month window, with more weight given to observations within each 6-month window. Thus, our final OHC estimates resolve OHC variability at roughly annual resolution.

**Description of the interpolation process.** The inputs to the ARANN include a set of basis functions that are used to approximate the spatial autocorrelations of the temperature anomalies. These basis functions consist of a set of sinusoids, which are used to approximate the correlation length scales found in the spatial maps of the temperature anomalies. To obtain these basis functions, we averaged the gridded temperature anomalies over the top 700 m for the years 2005-present and took the first 6 principal components of the resulting anomaly map, which explain >90% of the variance, in both the meridional and zonal directions. We also found that we obtained very similar principal components if we used the 700–2000 m layer instead of the 0–700 m layer. Using these principal components, we then estimated the periods of their autocorrelations, which then became the periods of our sinusoids. In the meridional direction, the six periods of the sinusoids are [360 180 90 60 45 30] degrees and in the zonal direction they are [180 120 90 60 45 22.5] degrees. Both the cosine and sine functions are used for each period, leading to a total of 24 basis functions. Due to how these sinusoids are constructed, they each represent a basis function that is merely a two-dimensional array of numbers between −1 and 1 with a characteristic length scale ranging from roughly 1000 to 18,000 km.

With these basis functions, the ARANN can reconstruct temperature anomalies on horizontal scales of roughly 500–1000 km. The ARANN is not designed to capture small-scale features and the OHC changes reconstructed by this method should be interpreted on scales of ~1000 km or larger. Indeed, when the ARANN is applied to temperature anomaly fields from global ocean models, the residuals

contain features on the order of several hundred km even where the spatial sampling is relatively dense and evenly distributed (see Supplementary Figs. 8–11). The presence of geophysical noise and sparse sampling further reduces the scales that we are able to resolve.

While the ARANN uses sinusoidal basis functions to capture spatial variability in the temperature anomalies, there are no explicit variables for time or depth. Instead, the ARANN uses a sweep from higher data coverage to lower data coverage in order to capture the depth and temporal dependencies (Supplementary Fig. 2a). This is done by slicing the three-dimensional temperature anomaly fields into two-dimensional "chunks" and applying a separate ANN to each consecutive two-dimensional temperature anomaly field, using temperature anomalies from previous depth- and time-slices as additional input features. Vertical mixing is important in certain regions and contributes to deep water formation, so surface warming of the ocean would be expected to display some imprint on the layers below. Because the ARANN is iterative, it optimizes for each depth interval during its sweep from the surface to seafloor. Relationships identified by the ARANN at depth will therefore evolve from those found at the surface. In this way, the ARANN mimics in a parameterized way the circulation and mixing processes that connect the surface and deeper layers of the ocean, and encodes some "memory" to capture the temporal evolution and persistence of temperature anomalies. The ARANN has limited memory, or what is sometimes called finite impulse, in that it does not retain the knowledge of all prior time-steps but only those that have occurred most recently. This simplification mimics the fact that the correlation between temperature anomalies at a given location in the ocean diminishes over time as heat is circulated and mixed away. Since the autocorrelations in the anomalies evolve over time, the ARANN refines and evolves the relationship between temperature anomalies and the input fields at each step.

The iterative process is described in detail below. At each step of this process, a new ARANN is developed, trained, and validated, following the procedure described in the next section.

1. Starting with the most recent 6-month period in the time-series (i.e., Jul–Dec 2019), we begin by randomly choosing the size of our initial depth window, which consists of between two and six depth layers. This corresponds to a depth interval of 10–150 m above 300 m. Starting with the depth window closest to the surface, we randomly select 50% of the data within that depth window for training the ARANN, setting aside the remaining 50% for validation (a similar 50/50 training/validation split is maintained through all subsequent steps). We then interpolate the temperature anomalies in this initial near-surface depth window using only the 24 sinusoidal basis functions as inputs to the ARANN. This is iterate (1,1) in Supplementary Fig. 2a.

2. Moving down to the next depth interval, we again choose a random depth window consisting of between two to six depth layers. The data sampling and interpolation within this depth window are repeated as in Step 1, using the interpolated temperature anomalies from the prior depth window as an additional input to another ARANN (again, a new ARANN is trained at each iteration). This is iterate (2,1) in Supplementary Fig. 2a.

3. Step 2 is repeated iteratively, moving down an additional depth level at each iterate, and using the interpolated temperature anomalies from all prior depth windows as additional inputs to the ARANN interpolation at each depth level. This means that at each new depth level, there are $24 + i$ inputs to the ARANN, where $i$ is an index corresponding to the number of vertical depth intervals used in the interpolation. Below 300 m, the depth window is expanded to between six and twelve depth layers (corresponding to a depth interval of 150–1200 m) to ensure that adequate amounts of data are available to the network. Randomizing the size of the depth window at each iteration ensures the model is not fixed to specific depth intervals and will eventually produce a smooth transition in the vertical gradient. Steps 1–3 complete the initial sweep over depth levels and fills in iterates (1,1) to (i,1) in Supplementary Fig. 2a. The total number of depth intervals used from the surface to the seafloor ranges from 11 to 27 intervals, depending on the (random) choice of depth layers used at each interval.

4. Steps 1–3 are then repeated iteratively, marching backwards in time at 6-month intervals through the time series. At each time step, interpolated temperature anomaly fields from six prior time intervals (or the maximum number available if less than six) are used as additional inputs to the ARANN. Using temperature anomalies from prior time intervals as additional inputs to the ARANN mimics the temporal autocorrelation of temperature anomalies and allows the propagation of information from well-sampled time periods to more sparsely sampled periods. We do not fix the autocorrelation timescale (other than limiting the inputs to 6 previous time steps, or 3 years), but rather let the network decide at each iteration how much weight to put on previous iterates when interpolating anomalies at each time step. At the conclusion of Step 4, we have 148 gap-filled three-dimensional datasets of temperature anomalies at 6-month resolution from 1946 to 2019.

5. After running the model backwards from 2019 to 1946, steps 1–4 are repeated, this time running time forwards from 1946 to 2019. In this forward sweep, we use interpolated temperature anomalies from three prior time intervals and three subsequent time intervals as additional inputs to the

ARANN at each time step. This forward sweep helps to further propagate information through the network and is particularly helpful for smoothing results from the transient stage at the beginning of the backward run when the model had less information on temporal autocorrelations. This is most important in the abyssal ocean where temporal autocorrelations are longer and data is sparser.

6. Steps 1–5 are repeated 10 times for each possible climatology, generating an ensemble of 60 ARANNs for interpolation, from which we can derive uncertainties related to data sampling, interpolation, and climatology.

7. Steps 1–6 are repeated four times, each time applying a different calibration method to the XBT and MBT data (see Supplementary Fig. 13).

An example of the resulting temperature anomaly field from this mapping method (Supplementary Fig. 3) reveals how our method takes the original binned temperature anomalies and interpolates the data to produce filled anomaly fields for a single realization of the ARANN. In the upper 50 m for the year 1960, the observational sampling neglects much of the southern hemisphere, but by leveraging information from prior iterations, the ARANN method produces a smooth product that fills gaps that would not be captured by traditional objective mapping. In the year 2010, the upper 50 m has much more regular sampling, and the ARANN method captures the large-scale patterns while smoothing over small-scale features. For 900–1100 m in the year 1960, temperatures are very sparsely sampled, but the ARANN method still produces large-scale regional structures that would not be captured by traditional objective mapping. The ARANN also smooths over most of the small-scale noise in the observations at these depths. This noise is quite apparent when considering the anomalies fields for 900–1100 m in the year 2010. The ARANN interpolation of this data captures the large, near basin-scale spatial patterns, while ignoring most mesoscale to sub-mesoscale patterns, which can be seen in the residuals (ARANN – Obs.) of the temperature anomalies.

**Architecture of the ARANN and method of solution.** Each step of the individual autoregressive artificial neural network (Supplementary Fig. 2b) consists of an input layer that contains 24 naïve basis functions (B), as well as filled temperature anomalies for six adjacent time steps (if available) ($\theta_t$), and anomalies for all prior depth intervals for the current time step ($\theta_z$). Each input is organized as a vector with a length $n$, equal to the number of spatial grid points in the gridded temperature anomaly fields. In total, there are $m$ input fields, where $m$ is between 24 and 46 (24 basis functions, 6 filled temperature anomaly fields from adjacent time-steps, and up to 16 filled temperature anomaly fields from prior depth intervals). These inputs are organized as an array $I$ with size ($n \times m$),

$$I = [B, \theta_t, \theta_z]. \qquad (1)$$

In the ARANN, the input "layer" connects to a single hidden "layer" with 10 nodes, producing a network with ($m \times 10$) input "weights" organized as an array ($W_1$). We selected this number of nodes by experimenting with adding more free parameters, the weights, until the performance of the ARANN on internal validation sets comprised of data sampled during periods of high data sparsity (pre-2005) no longer improved. The values for the hidden layer ($H$) produced by the ARANN are,

$$H = F(I \cdot W_1 + b_1), \qquad (2)$$

where $F$ is the "transfer function" used to propagate information in a non-linear fashion through the network, and $b_1$ is a [$1 \times 10$] array of "biases", with all values in a given column being identical. We use the hyperbolic tangent as the transfer function, which is commonly employed for interpolation[57–59] by fully connected feedforward networks like the ARANN. The output "layer" connects to the hidden layer using another [$10 \times 1$] array of weights $W_2$ and a single bias [$1 \times 1$] $b_2$ to produce the predicted temperature anomalies for the interpolation ($\theta_{intp}$),

$$\theta_{intp} = H \cdot W_2 + b_2. \qquad (3)$$

In all, each network has ($10 \times m$) + 21 free parameters, representing the ($10 \times m$) weights $W_1$ of the input layer, the 10 bias terms of the input layer, and the one bias term, and the 10 weights $W_2$ of the output layer. These free parameters are iteratively adjusted to achieve a minimum of a cost function that measures the mean sum of squares difference between the interpolated temperature anomalies ($\theta_{intp}$) and the observed temperature anomalies in our training dataset ($\theta_{obs}^{train}$),

$$cost = \frac{\sum_{k=1}^{N}(\theta_{intp}^k - \theta_{obs}^{train,k})^2}{n}, \qquad (4)$$

where $N$ is the number of observations within the training dataset at each iteration. As discussed in the prior section, the training dataset consists of a random 50% selection of all available data at each iteration. For the back-propagation algorithm in our network, which iteratively updates the values of the weights to minimize the cost function, we chose the Levenberg-Marquardt algorithm[60] due to its improved performance at reducing the error between predictions and observations versus other common algorithms such as gradient descent[61].

For each network, we withhold the 50% of data not selected for the training set as validation of the network. This validation dataset ($\theta_{obs}^{val}$), is used to prevent overfitting of the network, which occurs if the network is over-trained on a dataset

so that it cannot extrapolate well when presented with new data. We use an early stopping technique[62] to avoid overfitting, whereby the interpolated temperature anomalies are periodically checked against the validation dataset, and training is terminated when the root-mean-squared error of the interpolated temperature anomalies against the validation dataset begins to decrease.

**Required smoothness constraints.** After each time-step, we perform a further check on the interpolated temperature anomalies to ensure that they satisfy certain vertical and temporal smoothness conditions in their basin-averaged temperature anomalies. These checks are performed individually for the Atlantic, Pacific, Indian, and Southern Ocean basins (the Arctic is not considered for this check due to data sparsity) using information about the natural rates of change derived from observations during the Argo Era (defined as 2005 onwards). These checks are only performed for time-steps prior to 2005. The boundaries for these ocean basins are the same as defined in the main text and utilize the WOA 1-degree mask[63].

First, we calculate the change in basin-averaged temperature anomaly at each depth level from the current temporal iteration to the previous temporal iteration, $\Delta\theta_t(\text{basin}, z)$, from the interpolated temperature anomalies during the well-sampled Argo period (2005–2019). We also calculate the difference in basin-averaged temperature anomaly at each iteration from one depth level to the previous depth level, $\Delta\theta_z(\text{basin}, z)$, for the Argo period. This yields 28 values of $\Delta\theta_t$ and $\Delta\theta_z$ for each basin and depth level. We then require that $\Delta\theta_t(\text{basin}, z)$ and $\Delta\theta_z(\text{basin}, z)$ for the current iteration not exceed the 3rd standard deviation ($\sigma$) of $\Delta\theta_t(\text{basin}, z)$ and $\Delta\theta_z(\text{basin}, z)$ during the Argo period. That is,

$$\Delta\theta_t(\text{basin}, z)^{\text{current}} < 3\sigma[(\Delta\theta_t(\text{basin}, z))^{\text{Argo}}], \qquad (5)$$

$$\Delta\theta_z(\text{basin}, z)^{\text{current}} < 3\sigma[(\Delta\theta_z(\text{basin}, z))^{\text{Argo}}]. \qquad (6)$$

If a network does not pass both time and depth constraints for all ocean basins, then that network is rejected and the network must start over at the same time-step and depth interval. To allow for some additional flexibility, in the event that a network is rejected more than five times in a row, the run that produced the lowest average exceedance of these smoothing constraints is accepted and the procedure continues as usual. This is an uncommon occurrence, appearing only a few times in a single run from 2019 to 1946, but it most often occurs due to sharp changes in the temperature anomalies of the deep Pacific during the early 1970s.

**Statistical information.** 240 ARANN OHC members are used for all depth intervals in this study, representing 10 members for each combination of the four bias corrections[21–24] and six decadal climatologies[63]. All estimated warming rates for the periods specified in the text are calculated from the mean linear trend of the ARANN OHC members, and the uncertainty in these warming rates are calculated as 2 standard deviations across the individual linear fits for the ensemble members. All estimates in W m$^{-2}$ are for the entire surface area of the Earth. Total warming estimates in ZJ come from multiplying the warming rate derived from a linear fit by the length of the time period under consideration.

Error bars in Figs. 1, 2, and 4 represent 2 standard deviations computed from the ARANN ensemble members. In Fig. 3, the trends for each grid cell are considered significantly different from zero only if the trends exceed twice the sum of two components of uncertainty, 1. the ensemble-mean standard error of the linear fit, which is a measure of the amount of uncertainty due to short term variability in the warming rate at a given location, and 2. the standard error of the cross-ensemble linear trends, which is a measure of the uncertainty due to the ARANN mapping method.

For Fig. 5, warming rates for the specified periods are calculated from the mean linear trend of the ensemble and the error bars for the ARANN estimates represent the maximum and minimum warming rates after excluding the six members with the highest rates and the 6 with the lowest, representing the 95% confidence interval. Error bars in Fig. 5 for previously published warming rates come from the uncertainties calculated by these other studies, which may differ somewhat from each other due to methodological choices but roughly represent the 95% confidence interval.

## Data availability

The source data from the ARANN method underlying this study's Figs. 1–5 and Supplementary Figs. 3–15, as well as gridded temperature anomalies for the ensemble mean have been made freely available under accession code https://doi.org/10.6084/m9.figshare.12959489. Gridded temperature anomalies for individual ARANN ensemble members will be provided upon reasonable request. Raw temperature casts from the World Ocean Database are available at https://www.ncei.noaa.gov/access/world-ocean-database-select/dbsearch.html. Temperature climatologies from the World Ocean Atlas are available at https://www.nodc.noaa.gov/OC5/SELECT/woaselect/woaselect.html. Data used in this study from the IAP ocean heat content and IGOT temperature profiles are available at http://159.226.119.60/cheng/. Data from the JMA ocean heat content are available at https://www.data.jma.go.jp/gmd/kaiyou/english/ohc/ohc_data_en.html. Data from the NOAA ocean heat content are available at https://www.ncei.noaa.gov/access/global-ocean-heat-content/. Ocean heat content data from the Green's functions method are available at https://laurezanna.github.io/post/ohc_pnas_dataset/. Data from the optimized mixing model OPT-0015 are available at https://drive.google.com/file/d/

1dgpYPpGdt8fvr3aXvCFbUE9iyKnVVfAN/view. Data for the top of the atmosphere net radiative flux is available at https://researchdata.reading.ac.uk/111/. Data from the repeat hydrographic sampling is contained in Table 1 of https://doi.org/10.1002/2016GL070413. Additional data contained in this study's Fig. 5 can be obtained from https://doi.org/10.3389/fmars.2019.00432, https://doi.org/10.1038/nclimate3043, https://doi.org/10.1002/2014GL062669, and https://doi.org/10.1038/s41598-019-56490-z. Outputs from the MIROC and CNRM CIMP6 models can be obtained at https://pcmdi.llnl.gov/CMIP6/.

## Code availability

Code needed to implement the ARANN method and plot the source data contained in this study's Figs. 1–5 and Supplementary Figs. 3–15 have been made freely available under accession code https://doi.org/10.6084/m9.figshare.12959489. All code is provided "as is" and modifications may be necessary depending on users' available computational resources.

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

## Acknowledgements
T.D. acknowledges support from the National Science Foundation (OCE-1948985). Computing resources were administered by the Center for Scientific Computing (CSC) and funded by the National Science Foundation (CNS-1725797).

## Author contributions
A.B. and T.D. designed the study. A.B. performed the computations and analysis. A.B. and T.D. wrote the paper.

## Competing interests
The authors declare no competing interests.

## Additional information

**Peer review information** *Nature Communications* thanks the anonymous reviewers for their contributions to the peer review reports of this work. Peer review reports are available.

