## [Peer Review File · Nature Communications]

REVIEWER COMMENTS

Reviewer #1 (Remarks to the Author):

Bagnell and DeVries present a new estimate of post-1945 ocean heat content that has significant disagreements with CMIP-5 and community consensus. Such a disagreement is plausible because instrumental records before the satellite era contain considerable uncertainty that could be underestimated by previous studies. The manuscript has important findings, such as the changes in ocean heat uptake with variations approaching $\pm 1 \text{ W m}^{-2}$ on interannual timescales and a recognition that the deep ocean has greater variability than suggested by a Green's function method. The manuscript has problems related to communication, including the title. Recent acceleration in Earth's energy imbalance is well known, and the story here is that the energy imbalance didn't begin until much later than the community consensus. In addition, the method should be placed into context more carefully. The manuscript is interesting in that it takes a new neural network algorithm, it does a good job validating the model against artificial data, and it comes up with a result that questions the status quo. The complexity and novelty of the method, however, undermines confidence that the algorithm accurately reflects the truth. A convincing case would demonstrate that this method overcomes some obstacle that other methods cannot. Evidence of what went wrong with other estimates is part of this case. Without such evidence, the reigning null hypothesis is that all ocean heat content estimates prior to the satellite era are actually consistent with each other due to their underestimation of the true uncertainty.

Specific points:

L11: The ocean is presently the dominant energy sink, but the cryosphere was similarly large in the past.

L15: Not true, "the first global reconstruction of historical changes in top-to-bottom ocean heat content based on in-situ temperature data". Zanna et al. 2019 is another estimate based on in-situ observations. Perhaps the authors mean that they use raw or point-wise observations? This is quite a different statement than the one in the abstract.

L18: What does "effectively" mean? How does it compare to interpolation using a dynamical model (e.g., Zanna et al. 2019)?

L22: Obviously the energy balance is not perfect, so when the authors claim a balance they must mean within some threshold.

L25-27: The finding that ocean heat uptake before 1990 was negligible is plausible to this reviewer, but is contrary to community consensus.

L41: To imply that objective mapping is the right way to infer deep ocean heat content is misleading. Furthermore, it may not be the method that is the problem; it could be the sparsity of data below 2 km depth. Later in the paragraph, the discussion of dynamical-model interpolations is valuable.

L84: The switch of the deep ocean from cooling to warming is nearly identical to that found by Gebbie (2021, *Ann. Rev. Mar. Sci.*, Figure 6).

L93: "data-based" is a vague description that can be defined in many different ways. Most readers will consider this statement to be false.

Figure 1: Consider using ZJ for easier comparison to other studies. The y-axis labels in A and B are inconsistent. ORAS4 goes off the scale in D. The figure is important and could have greater resolution.

Figure 1: The GF method has more ocean heat uptake from 1950-1980 and appears to be an outlier here. However, other estimates from Domingues et al. 2008, Ishii et al. 2017, Cheng et al. 2019, and Gebbie & Huybers 2019 also have a character similar to the GF method in the 1950-1980 interval. That doesn't mean that they are correct, of course, but that this manuscript represents an overturning of the community consensus.

L110: Between 2005-2018, the RAPID-MOCHA data do not suggest any slowdown in AMOC. A decrease up to 2013 has been counterbalanced by recent events.

L612: It is a real judgement call to categorize the Green's function method as being "model-based". It uses a circulation based on data ingested in an ocean reanalysis, and it uses data for boundary conditions. I believe it is unnecessary and potentially misleading to call the Green's function method "model-based".

L613: "and and"

Figure 4: Why should a flux be called "ocean heat storage"? This terminology seems incorrect.

Figure 5: ARANN reports that ocean heat uptake is known within $\pm 0.02 \text{ W m}^{-2}$ for the interval 1960-2018. How is this possible with the sparsity of data before the Argo, WOCE, and satellite eras?

L180: "they also suggest that previous estimates of global warming may have been biased too high prior to 1990 due to the neglect of deep ocean cooling during this time period." Why would this be the case? If a reason can be found, that would elevate the credibility of ARANN above the other estimates.

ARANN consistently produces low values of ocean heat uptake relative to other estimates. The 6-month averaging scheme is essentially a fall and spring average. Perhaps the extreme seasons are not being accounted for in the same way as other estimates. It would be highly illuminating to check the sensitivity of this 6-month timescale.

L329: It is a concern that 30 or 90 or 120 realizations (different numbers are cited) are not enough to capture the error covariance, potentially leading to error bars that are too small.

Reviewer #2 (Remarks to the Author):

The paper 'Closure of Ocean Heat Budget Reveals Recent Acceleration in Earth's Energy Imbalance' provides insight into global scale ocean warming, and rate of change linked to the Earth energy imbalance (EEI) based on a new product ARANN allowing to address full-depth ocean change. The development of the new product is an important contribution on climate science, and particularly their new results allow further discussions on changes of the EEI. Aside the very well written and presented method of the new product, its evaluation in the context of climate studies lacks robustness, and in-depth validation. ARANN global estimates disagree to previously published products and results, and their robustness thus demands rigorous in-depth validation, which is not provided. Moreover, some of the outcomes have been linked to ocean processes and climate modes of variability without any further analysis steps. I therefore recommend rejection, and resubmission to another (more technical) journal for the new method after major revision. Further comments below.

L.11: The ocean is a dominant heat sink – could be also either energy – mostly in the form of heat ...

L.11-12: The Ocean sink is about 90%, so to be correct, add 'recorded predominantly by' ...

L. 40: too generic message on 'sparse measurements', and does not recognize the unique improvements of the observing system component with the implementation of Argo. Without this, science advances on the EEI could not have been made. This needs corresponding recognition in the revision of this sentence.

L.76-84: These results contradict previous results, and Earth heat gain appears to be much lower than estimates elsewhere. Why is this the case? Why does this not occur in the paper? There is no critical reflection on the outcome from this new product compared with what has been published until now, and needs to be added.

L.78-79: Fig. 1A shows an anomaly, and it is not straightforward to re-plot the values for heat accumulation provided by the authors for both, the fluctuation, and the accumulation since 1990. Moreover, I am also puzzled by the strong-year-to-year variations at the end of the time series in ORAS4, which is not reproduced by the other time series (eg from 2010), what is happening there?

L. 85-87: The figures do not show a 'broad agreement' at all, and already the results presented in L.76-84 do not provide 'broad agreement'. Or what is intended to say with such a very general statement on 'broad agreement'? This is not clear. Moreover, a fundamental discussion is missing on the criteria for the choice of products used to compare the results from the new method. Why is IAP chosen? Why is ORAS4 chosen (by the way a more updated version ORAS5 already exists)? Particularly for the latter, this appears to be critical with respect to the statement the authors provide earlier in the manuscript (L.-47-49) – With respect to the information provided by the authors earlier in the text: why is now this reanalysis the one to be used as a critical reference for demonstrating a 'broad agreement'? Since 2010 for example, i.e. when data sampling is high in the upper 2000m depth from Argo, ORAS4 shows much higher values than the other time series do. How is this explained? This discussion is much too vague.

L.88: Fig. 1B indicates that the new method disagrees strongly with what has been obtained in IAP, and almost does not show any warming prior to 1990 – that is what had been discussed earlier (L.76-84) and this statement appears to contradict and not be consistent.

L. 89: It is challenging to see an agreement for an 'oscillatory behavior' between ARANN and ORAS4? How do these time series correlate? Very vague statement for a result validation discussion.

L.90-91: This statement appears to be not founded on what is shown in Fig. 1C – ARANN does not show any warming trend – at least what can be qualitatively derived from the figure (no trend values available to further found this discussion?) – whereas IAP does show a clear warming trend. Similar to my comment above this contradicts to what has been written in the paragraph above (L.76-84), and is inconsistent in itself.

L.93-95: Results of Zanna et al. (2019, cited in this study) should be discussed here in this context, and relativize this statement accordingly (although shown in Fig. 1). It would be good to understand for example whether this cooling had been reported elsewhere? Moreover, a discussion is missing on why the Zanna et al. do not show a similar signal? Moreover, as stated above, the comparison to an ocean reanalysis has limitations as stated by the authors in the introduction part. And this caution is further strengthened by looking at Fig. 2, in which the reanalysis result appears to strongly disagree? Additionally, no trends are shown in Fig. 1, only the time series, although the change of trends over time are the major tool of argument for this qualitative comparison/discussion.

L.95-96: Given that the layer above does not really show a warming signal, I am a bit puzzled about this statement. Has this been somehow quantified? This is not convincing from what is shown in Fig. 1 only. – particularly given that this is also highlighted as one of the major outcomes in the abstract (L.21)?

L.105-107: Could the authors provide the analysis for the statement that the observed warming (or as shown the OHC anomaly) is consistent with vertical heat penetration from AMOC? This is not clear. Also not further reference is provide, nor a quantification to support this statement. Vertical penetration of heat is also not shown with Fig. 2.

L110-116: These statements based on Fig. 2 time series are an over-interpretation, and there is still major discussion on how or whether water mass formation with AMOC variability (eg Buckley and Marshall, 2016, doi:10.1002/2015RG000493). In addition, there is no study available which provides evidence on the use of a basin-wide OHC estimate as an indicator for AMOC variability and change. The link of upper ocean heat changes with AMOC variability has been discussed elsewhere (eg Zhang and Yang, 2017: <https://www.nature.com/articles/s41598-017-14158-6>), but specifically focused and analyzed to the processes. If a link to AMOC is thought to be established, a more precise analysis has to be performed. Likewise, there is still no consensus about the 'warming hole' and AMOC, and in my point of view this should be mirrored in the text, particularly when aiming to publish in a journal for a wider audience – see chapter 5 in IPCC SROCC for example on more content for this discussion (<https://www.ipcc.ch/srocc/>).

L.117-125: Recommend to include recent publication into this discussion:
<https://www.nature.com/articles/s41467-020-15754-3>

L.122-124: see comment above – a more precise analysis is needed to connect the basin-wide OHC time series to such process discussions.

L.129-132: Also here, more analysis is needed for such as statement, and the link to climate variability indicators such as for example to the PDO. With this discussion, the link is not established. An example of how this could be done qualitatively is in Dieng et al., 2017, their figure 6 (DOI: 10.1002/joc.4996).

L.136-138: there is a recent publication linking to the paleo context and could support further discussions: <https://www.pnas.org/content/116/30/14881>. However, the cooling signals as shown in ARANN are not reflected in the other time series, any explanation?

L.102-147 & Fig. 2: I am surprised to see no mention on the huge differences between the ARANN results and the other time series which have been claimed to be under 'broad agreement' in Fig. 1 – how can this be explained given that particularly the ocean reanalysis shows very different results – and which had been discussed as a further confirmation on ARANN deep ocean changes in the paragraph above? Why do these differences occur, and how can they be interpreted in the context of this discussion?

L.151-153: I would be curious to see the results obtained for $d(\text{OHC})/dt$ using the other products for comparison. How different are they? Do they show a similar result? And additionally, a time series for net flux at TOA should be also included to comparison to further analyze the robustness of the results. The CERES product for the most recent period, and a long reconstruction such as for example from Allan et al. 2014, <https://agupubs.onlinelibrary.wiley.com/doi/full/10.1002/2014GL060962>. In addition – this is what would be required to 'close the budget of the Earth energy imbalance' – this is what the title promises – but the entire paper does not provide any analysis step of discussion linking to 'budget closure'.

L.161-164: Also as discussed above, there is further evaluation needed: why not adding the climate mode indicators to the figure to further underline the discussion? Neg./pos. EEI values are not news knowledge, but a more in-depth attribution study would be of strong value for climate science. Also, the evolution for 'El Chichon' shows an increase towards 0 EEI. How can this be interpreted?

L172-176: This is not the case, see discussions above, and the authors own statements. Moreover,

such a discussion would only be valid if the other results would have been added into Fig. 4 as well.

L.180-182: Yes, all other products / method disagrees with early ARANN estimates. Why is this the case? If it is due to the fact that ARANN provides an improved estimate for deep ocean changes, why is this not further validated, either through comparison to regional (if available) deep ocean measurements, or under a physical budget approach (e.g. by comparing with net flux at TOA ?), or the sea level budget while using reconstructions, etc.? A more robust validation of the results are needed to assure robustness of the outcomes.

L.189-192: This statement needs more clarification, and it is not clear on how the 3 volcanic eruptions induce a 'lack of ocean warming' – what does this mean? And how can a climate forcing that acts on short (1-2 years max) counteract a signal over the period 1950-1990, thus several decades? Please further clarify.

L.200-203: This statement needs also further explanation to understand the main message, this is very general, and not clear.

L.231: Not clear on the use of the climatology: Using only a most recent climatology, which is comparable warm (ie on the top of the strong warming observed during the past decade) could potentially explain why a low (or no trend) is observed at the beginning of the time series? How robust is the method compared to different climatologies (eg. starting with a 'cool climatology' for 1960-1970? And one 1970-1990, and then one 1960-2018?)? Would this effect the global trends of ARANN?

Reviewer #3 (Remarks to the Author):

Review of: Closure of Ocean Heat Budget Reveals Recent Acceleration in Earth's Energy Imbalance

Authors: A. Bagnell and T. DeVries

This study constructs a state-of-the-art estimate of historical ocean warming between 1946-2018. This reconstruction is unique in that it covers the entire water column (full depth), and thus includes the relatively large volume of the global ocean that is generally unaccounted for in estimates of global energy imbalance. The authors do so by applying novel machine learning techniques to populate regions of sparse data coverage, which is a particularly significant accomplishment in the deep and abyssal ocean where data coverage is quite limited is notoriously difficult to constrain. The application of this methodology to ocean heat content estimates is new and performs quite well. The authors find that the addition of estimated abyssal heat uptake implies very little total imbalance in TOA radiation between 1950-1990, which conflicts with other (less constrained) estimates, a result that will interest a broad range of climate and ocean scientists. I really enjoyed reading this paper and think it would be appropriate for Nature Communications, both for its relevance to a number of fields as well as the implications of the findings.

I have one technical issue and one general point the interpretation of results, as well as several minor questions/comments I'll list below.

Technical issue:

Lines 231-253: I am having trouble understanding why the choice of seasonal climatology, and specifically the use of a climatology from the latter part of the record (2005-2017) doesn't bias the

result. I see that you do explore the sensitivity of results to the choice of climatology, concluding that it doesn't matter much, but I am confused about how this could be. If I'm understanding correctly, ocean warming as presented in the paper (i.e., Figs. 1-5), is defined as the anomaly between the data and this seasonal composite. If so, aren't you subtracting a climatology that has a higher average temperature, given the fact that it was calculated using data in which the ocean has warmed substantially from earlier in the record (at least in some regions). I list this concern first because the negative heat content anomaly earlier in the record (1950-1990), particularly in the deep ocean, is an important result of the paper; I am having trouble understanding how this isn't a unavoidable artifact of your methods.

I am likely missing something, but I would be more convinced if you were to provide a more physical explanation for the insensitivity to the seasonal climatology. Further, I wonder if the equivalent to Fig. S14 could be performed with one the models used in the study, given the issues with the observational data coverage from the earlier climatology that Fig. S14 is based upon. However, if there is a simple explanation for the insensitivity to the use of climatology, further analysis isn't necessary. Regardless, a clear statement about how temperature anomalies were defined should be in the main manuscript (somewhere between L39-55).

Interpretation issues:

I found a few areas where the paper would be enriched by clearer interpretation of the results.

Most generally, and given the emphasis of the paper, I feel the work could be strengthened by a more concrete comparison between the EEI presented here to estimates of radiative forcing. Specifically, the results in Fig. 5 and discussed in the manuscript imply that previous estimates of energy imbalance in the climate system over much of the historical record have been biased high. The authors discuss in (lines 184-206) that this finding is at odds with estimates of positive radiative forcing. It would be interesting to see these radiative forcing estimates on Fig. 5, giving a more quantitative understanding of how significantly variability can influence the energy budget. Also, are the cooling estimates here at all consistent with Gebbie & Hubers (GRL, 2019)? It would be good to include discussion of the relationship of deep temperature anomalies to historical surface temperature anomalies in ventilation regions, in addition to the discussion of variability.

Likes 102-116: I found myself re-reading this passage multiple times to understand your points. I think it is confusing because you have partitioned the data into two epochs in time, and several layers in depth, and skip back and forth between them in this discussion. It would help if you were more consistent in summarizing results around the two epochs in time you use throughout the paper. I'm also confused with your interpretation: you attribute the deep Atlantic transition from cool to warm anomalies to a possible slowing of the AMOC after 2005, but also attribute the dipole in upper ocean temperatures prior to 1990 to a cessation of deep-water formation. Are you not attributing two different features to the same phenomenon, but at different and inconsistent times?

The same comment, regarding being more clear/consistent about the time periods compared, applies to the following few paragraphs on regional results (117-147).

Lines 159-162: Have you compared your results to an ENSO or IPO index? If a relationship existed there, it would be a nice addition.

Lines 200-204: Is it possible to place the magnitude of variability in heat content found here to other estimates? Even, perhaps, the models explored the study (despite their flaws in deep ocean physics and water mass properties). Or historical records of deep ocean temperatures? I am left wondering how the findings here compare to other estimates, though I realize one of the points of the paper is that similar estimates are lacking from the literature. However, even a qualitative comparison of your results to others would be

Small comments:

Line 117: Define what you're calling the Southern Ocean (in fact, the same applies to Atlantic and Indo-Pacific). I assume all regions are partitioned at 30S?

Line 163: This sentence is a bit misleading. Variability is still important later in the record, it just doesn't override forcing. Please reword.

Line 165: You discuss this as though the consistency is a confirmation of the methods, but don't Fig. 1 and Figs. 4-5 have to be consistent by construction (aren't you just taking the derivative of OHC from the same dataset)?

Line 177 and Figure 5: Please note in the caption that the time periods averaged over in Figure 5 are different from the previous figures. I was confused for a bit about how Fig. 4 and Fig. 5 weren't at odds.

Line 199: add "the" year 2000.

Line 257: what do you mean by the characteristic length scale in this context? Of the sinusoids?

Line 265: here you refer to the horizontal spatial variability, correct? If so, add the word "horizontal" or "lateral" or something.

Line 272:274: This is an interesting point. Does this provide an opportunity for physics-based discovery, specifically, as an independent diagnostic of implied timescales of ocean memory?

Related to this point, I have some confusion around lines 283-291. I'm somewhat confused about the iterative incorporation of additional vertical layers. Does it not matter that characteristic scales of vertical gradients in, say temperature, will differ between shallow and deep regions of the ocean? Should we expect an algorithm training on shallower data to be applicable deeper in the ocean? Could you please expand here on why you incorporate progressively deep data in this way?

Fig. S2A Caption: "near the."

Reviewer #1

Bagnell and DeVries present a new estimate of post-1945 ocean heat content that has significant disagreements with CMIP-5 and community consensus. Such a disagreement is plausible because instrumental records before the satellite era contain considerable uncertainty that could be underestimated by previous studies.

It is true that we find less warming over the 1945-1990 period than most other reconstructions, due in large part to deep ocean cooling that has not been resolved in previous studies. Nonetheless, our findings do not disagree with the community consensus outside the uncertainty bounds provided by our analysis and by previous analyses. To make this point clear in the paper, we have included comparison with additional products in Figures 1, 2, and 5. In particular Figure 5 shows the agreement within error bars of warming rates estimated by the ARANN and different methods. We have also provided Fig. R1 at the end of this document for the reviewer, which presents the warming rates estimated by the IPCC's SROCC Table 5.1 versus those estimated by ARANN.

The manuscript has important findings, such as the changes in ocean heat uptake with variations approaching $\pm 1 \text{ W m}^{-2}$ on interannual timescales and a recognition that the deep ocean has greater variability than suggested by a Green's function method.

Thank you.

The manuscript has problems related to communication, including the title. Recent acceleration in Earth's energy imbalance is well known, and the story here is that the energy imbalance didn't begin until much later than the community consensus.

Thank you for raising this point. In response, we have changed the title to "Closure of Ocean Heat Budget Reveals Delayed Acceleration of Earth's Energy Balance".

In addition, the method should be placed into context more carefully. The manuscript is interesting in that it takes a new neural network algorithm, it does a good job validating the model against artificial data, and it comes up with a result that questions the status quo. The complexity and novelty of the method, however, undermines confidence that the algorithm accurately reflects the truth.

We agree that this is a novel method. That is why we have provided extensive validation of the method using ocean models in the Supplementary Material. As the revised manuscript says: "We have tested the method extensively on datasets from two ocean models used in the Climate Model Intercomparison Project Phase 6 (CMIP6), demonstrating the ability to accurately reconstruct OHC changes on both global and basin scales (Supplementary Figs. 4-7) at all depths of the ocean, and to recreate modeled temperature anomalies at spatial scales of ~ 1000 km or larger (Supplementary Figs. 8-11), even in the presence of realistic geophysical noise that is present in the observations but not the models (Supplementary Figs. 4-12)."

A convincing case would demonstrate that this method overcomes some obstacle that other methods cannot. Evidence of what went wrong with other estimates is part of this case. Without such evidence, the reigning null hypothesis is that all ocean heat content estimates prior to the satellite era are actually consistent with each other due to their underestimation of the true uncertainty.

The reviewer's null hypothesis is correct, as discussed in the manuscript and as shown in the revised Figure 5. The revised Figures 1 and 2 also provided more comparison with previous reconstructions. The takeaway is that our new reconstruction finds slightly less cooling than previous mapping methods in the upper 2000 m, along with the new finding of deep ocean cooling prior to 2000. But the uncertainties overlap with the range from other studies.

While we think that showing where previous methods "went wrong" is certainly beyond the scope of this paper (nor would we want to write such a paper), we agree that there is a need for more intercomparison with other products. The revised manuscript now provides extensive comparison and discussion of the similarities and differences between our OHC estimation and previous products, such as objective mapping products as well as some additional deep ocean datasets that were not considered previously. The revised Figures 1 and 2, as well as Figures 4 and 5, provide this additional comparison, as well as associated discussion.

To more directly address the reviewer's curiosity about the differences between our reconstruction and previous objective mapping products, we offer the following considerations: (i) Objective mapping methods differ somewhat from each other, but generally do not account for factors that we have considered in our study, namely climatological choice and different corrections for instrumental biases. Additionally, the impact of the underlying temperature dataset, quality control, and vertical resolution of the grid used in the mapping procedure are factors that have not usually been considered in other studies or in the current study (they are beyond the scope of this study). Nevertheless, we conclude that our estimate for OHC in the top 700 m is in line (within uncertainty) with objective mapping methods, and that differences in estimates for the 700-2000 m interval may arise partly due to the version of ocean cast data used, as the ARANN reconstruction agrees better with the IAP estimate at this depth interval when using the IGOT dataset published by Cheng et al (2016). See Fig. R2 at the end of this document. We believe that the WOD18 dataset is the most up-to-date dataset and have used this dataset along with the quality-control procedures discussed in the manuscript. A rigorous intercomparison of different datasets is beyond the scope of this study. We merely provide the figure for the reviewer's reference.

(ii) Traditional mapping methods are unable to cover depths below 2000 m, so this is where the complexity of the ARANN method is mainly justified. Covering depths below 2000 m is an obstacle that ARANN overcomes that other mapping methods cannot, as evidenced by the ARANN reconstructions of CMIP6 derived OHC change in the 2000-5500 m depth interval (Fig. S4-7). Recent studies such as Cheng et al. 2019, neglect the impact of the deep ocean for years prior to 1985 on global OHC change. Therefore, establishing whether a significant trend in the deep ocean occurs prior to 1985 is necessary to determine the resulting EEI. With the support of additional deep ocean datasets, we conclude that there are significant trends in the deep ocean that the ARANN is able to capture. In fact, cooling of the deep ocean prior to 1990 does appear to compensate for some minor warming of the top 700 m over the same time period. This is

supported by a passive transport method from Gebbie and Huybers 2019 that considers longer time-scales than the GF method of Zanna et al., emphasizing the need to account for long-term climate modes in such products.

Specific points:

L11: The ocean is presently the dominant energy sink, but the cryosphere was similarly large in the past.

Thank you for pointing this out. We now say that " The ocean is *presently* the dominant energy sink in the Earth system".

L15: Not true, "the first global reconstruction of historical changes in top-to-bottom ocean heat content based on in-situ temperature data". Zanna et al. 2019 is another estimate based on in-situ observations. Perhaps the authors mean that they use raw or point-wise observations? This is quite a different statement than the one in the abstract.

The Zanna method relies on SSTs that have been objectively mapped and a steady-state circulation model constrained by in-situ data, but this differs significantly from an approach that seeks to directly in-fill subsurface temperature data. We clarified our statement to indicate that we sought to do the latter, as follows "Here, we provide the first global reconstruction of historical changes in top-to-bottom ocean heat content based on *interpolated subsurface temperature data*".

L18: What does "effectively" mean? How does it compare to interpolation using a dynamical model (e.g., Zanna et al. 2019)?

We were referring to the ability of the ARANN method to interpolate in data-sparse regions (e.g. Supplementary Figs. 8-11). The method from Zanna et al. was not designed to produce such spatial reconstructions and has not been constrained by an optimization routine to recreate subsurface anomaly fields, unlike data assimilated circulation models and objective mapping methods. Instead, the Zanna method averages SSTs over large spatial patches (see their Fig. 2), so is not an interpolation method per se.

L22: Obviously the energy balance is not perfect, so when the authors claim a balance they must mean within some threshold.

That's true. The revised manuscript says " a roughly balanced Earth energy budget within -0.16 to 0.06 W m⁻² over most of the latter half of the 20th century".

L25-27: The finding that ocean heat uptake before 1990 was negligible is plausible to this reviewer, but is contrary to community consensus.

The findings here fall within the uncertainty range of prior studies (refer to error bars of updated Fig. 5), though certainly on the lower range, so this does not stray too far from the community

consensus. Note, however, that the community consensus does not currently include the deep ocean prior to 1990. This study shows that deep ocean cooling has contributed to the overall negligible ocean heat gain prior to 1990.

L41: To imply that objective mapping is the right way to infer deep ocean heat content is misleading. Furthermore, it may not be the method that is the problem; it could be the sparsity of data below 2 km depth. Later in the paragraph, the discussion of dynamical-model interpolations is valuable.

You're right, the manuscript was not clear about this. Current objective mapping methods are not the right way to infer deep ocean heat content. The reviewer is correct that much greater sparsity below 2000 m is a hindrance to any method, including objective mapping. We have clarified this, as objective mapping is not currently considered a feasible approach to estimating the deep ocean heat content, leaving only dynamical models, passive heat transport models, and the method laid out in the current study. Please see the 3rd paragraph in Introduction of the revised manuscript.

L84: The switch of the deep ocean from cooling to warming is nearly identical to that found by Gebbie (2021, Ann. Rev. Mar. Sci., Figure 6).

Thank you for pointing this out. We have now included the results of Gebbie and Huybers (2019, Science) in our updated Figures 1 and 2. There does seem to be some broad agreement here, although the timing differs somewhat in terms of the shift from cooling to warming. Prior to 1990, we find that there is good agreement between their method and ours for the cooling trend in the Deep Pacific and Indian. Overall, passive transport methods provide a useful semi-independent estimate of OHC change, but will miss the OHC changes that are driven by ocean circulation changes.

L93: "data-based" is a vague description that can be defined in many different ways. Most readers will consider this statement to be false.

You're right, "data-based" is too vague. We have updated our terminology here, as we really mean direct in-filling of the observed sub-surface temperature anomalies in the spirit of objective mapping approaches.

Figure 1: Consider using ZJ for easier comparison to other studies. The y-axis labels in A and B are inconsistent. ORAS4 goes off the scale in D. The figure is important and could have greater resolution.

We have updated the units to ZJ, and created additional tick marks to allow for easier reading of the figure. We have also included additional OHC estimates from 3 different objective mapping products in the upper 2000 m, and the methods of Zanna et al. and Gebbie and Huybers in the deep ocean, as well as repeat hydrographic observations in the deep ocean. To make room for all of these comparisons we removed the comparison to ORAS4. We hope the resolution looks okay as it looks good on our end.

Figure 1: The GF method has more ocean heat uptake from 1950-1980 and appears to be

an outlier here. However, other estimates from Domingues et al. 2008, Ishii et al. 2017, Cheng et al. 2019, and Gebbie & Huybers 2019 also have a character similar to the GF method in the 1950-1980 interval. That doesn't mean that they are correct, of course, but that this manuscript represents an overturning of the community consensus.

We have updated Figure 1 as described above. The GF method and Gebbie and Huybers differ from our OHC estimate and each other from 1990 onwards, whereas the objective mapping products agree well with our method over the 0-2000 m interval for 1990 onwards. Additionally, we mostly agree with the warming trends derived from objective mapping methods (as well as CMIP5 climate models) for the earlier period within the respective uncertainties of the various methods, as demonstrated in the updated Figure 5. Also, one of the important points of our study is to close the ocean heat budget by including the deep ocean cooling prior to 1990, which reduces the warming in that interval. In all, we don't think that the results presented here represent an overturning of the community consensus, but they will help to adjust that consensus to lower values in the period prior to 1990.

L110: Between 2005-2018, the RAPID-MOCHA data do not suggest any slowdown in AMOC. A decrease up to 2013 has been counterbalanced by recent events.

You're right, and there is an issue of comparison across different timecales here. What we want to highlight is simply that the fingerprints of the AMOC can be found in the OHC trends in the North Atlantic. We have revised and clarified this point as follows: "Fingerprints of ocean circulation changes are also apparent in the spatial distribution of warming rates (Fig. 3). Prior to 1990, the upper 700 m of the subpolar and polar North Atlantic was cooling, while the Gulf Stream extension region was warming (Fig. 3a), consistent with trends that have previously been identified as fingerprints of reduced deep convection in high-latitude deep water formation regions^{28,31}. Since 1990 there has been coherent strong warming throughout most of the Atlantic basin, except for a small patch of cooling in the center of the North Atlantic subpolar gyre (Fig. 3d). Warming of the North Atlantic from 1990-2005 has previously been linked to a surge in the Atlantic Meridional Overturning Circulation (AMOC) after 1990^{32,33}, although this has been followed by a decline in the AMOC after 2005 and cooling of the subpolar gyre^{33,34}, potentially contributing to the cooling trend identified over this longer period in the central subpolar gyre (Fig. 3d)"

L612: It is a real judgement call to categorize the Green's function method as being "model-based". It uses a circulation based on data ingested in an ocean reanalysis, and it uses data for boundary conditions. I believe it is unnecessary and potentially misleading to call the Green's function method "model-based".

Yes, that is true. We have updated our terminology to refer to the Green Function method of Zanna et al., and the mixing model method of Gebbie and Huybers, as "passive transport methods". We clarify in the introduction that these methods propagate observed sea surface temperature anomalies in to the interior ocean using steady-state ocean circulation (Intro, paragraph 3).

L613: "and and"

Revised

Figure 4: Why should a flux be called "ocean heat storage"? This terminology seems incorrect.

This is terminology conceived by others (see Meyssignac et al. 2019). OHC Tendency has also been used (see Trenberth 2014). Something clearer and almost as succinct is to call it $dOHC/dt$, as this is really what it is, the time derivative of the OHC, so we have updated our terminology accordingly.

Figure 5: ARANN reports that ocean heat uptake is known within $\pm 0.02 \text{ W m}^{-2}$ for the interval 1960-2018. How is this possible with the sparsity of data before the Argo, WOCE, and satellite eras?

Our uncertainty is somewhat larger than ± 0.02 (about ± 0.05 in the revised manuscript), but this is the result of calculating a linear trend for the warming rates. The longer the trend the more robust it is and the smaller the error bars, which is why they are smaller than shorter periods like 1990-2019 and 2000-2019. The published uncertainty of Zanna et al. and of Cheng et al. (the error bars in Fig 5) are similarly small. Various methods can be employed to calculate uncertainty, some of which could increase our error bars, but since we mostly agree with other methods within the uncertainty of present error bars (updated Fig. 5), using an established method of calculating our uncertainty, the current error bars are likely loose enough.

L180: "they also suggest that previous estimates of global warming may have been biased too high prior to 1990 due to the neglect of deep ocean cooling during this time period." Why would this be the case? If a reason can be found, that would elevate the credibility of ARANN above the other estimates.

Some other studies (e.g. Cheng et al. 2019) have explicitly assumed that warming/ cooling before 1990 in the deep ocean was negligible. We contend that this is not the case, as the deep ocean is not actually in steady-state and, based on our findings, exhibits a non-negligible cooling trend for part of the record. This would have to be due to a consistent negative forcing some time centuries ago, as it could take that long for the signal to propagate into the deep ocean. Gebbie and Huybers propose the Little Ice Age as a possible reason, and our trends agree well with theirs in the deep Pacific and Indian. This is discussed in the revised manuscript.

ARANN consistently produces low values of ocean heat uptake relative to other estimates. The 6-month averaging scheme is essentially a fall and spring average. Perhaps the extreme seasons are not being accounted for in the same way as other estimates. It would be highly illuminating to check the sensitivity of this 6-month timescale.

This is a good idea. We have updated our analysis to use an equal number of realizations with a six-month binning that average Jan-Jun and Jul-Dec as well as Apr-Sep and Oct-Mar (so winter/summer-centered and also spring/fall-centered). In the end, we found that this made a negligible difference, but we have kept these different binnings in our final ensemble. We have

also updated our analysis to utilize 6 different decadal climatologies [1955-1964 1965-1974 1975-1984 1985-1994 1995-2004 2005-2017] from the World Ocean Atlas, in order to more fully account for uncertainty in the climatology. So our revised OHC estimate is now based on 240 ensemble members in all.

L329: It is a concern that 30 or 90 or 120 realizations (different numbers are cited) are not enough to capture the error covariance, potentially leading to error bars that are too small.

Our updated analysis now consistently uses a total of 240 ensemble members for all uncertainty calculations. These different members account for uncertainty in binning procedure (winter/summer-centered vs. spring/fall-centered), in choice of climatology (one of 5 different climatologies), in BT bias correction (one of 4 correction methods), and in data distribution (using a randomly-selected 50:50 split of data into training and validation).

Reviewer #2

The paper ‘Closure of Ocean Heat Budget Reveals Recent Acceleration in Earth’s Energy Imbalance’ provides insight into global scale ocean warming, and rate of change linked to the Earth energy imbalance (EEI) based on a new product ARANN allowing to address full-depth ocean change. The development of the new product is an important contribution on climate science, and particularly their new results allow further discussions on changes of the EEI.

Thank you.

Aside the very well written and presented method of the new product, its evaluation in the context of climate studies lacks robustness, and in-depth validation.

The Supplemental Material provides in-depth validation of our method by applying it to temperature fields from ocean models, showing the ability to recreate modeled temperature fields from the observed data sparsity on global to basin scales at all depth intervals, even in the presence of realistic geophysical noise. So the method is fully validated and demonstrated to be robust.

We think what the reviewer might mean is that there should be more comparison to previous OHC estimates and discussion thereof. We wouldn't refer to this as validation, but we do agree with the reviewer that it is important to assess the degree of agreement/disagreement between the present study and previous studies. To address this, we have now added comparison with multiple objective mapping products as well as deep ocean datasets (see revised Figures 1, 2, and 5). These additions should bolster our conclusions about the nature of the EEI over recent decades, including impacts from changes in the deep/ abyssal ocean, and better place our study in the context of the current state of knowledge regarding OHC change over the past 70 years.

ARANN global estimates disagree to previously published products and results, and their robustness thus demands rigorous in-depth validation, which is not provided.

As mentioned above we provide rigorous in-depth validation of the method in the Supplemental Material. The revised manuscript now also provides a thorough comparison between the ARANN OHC estimates and those of previous methods. We find that the ARANN produces slightly less warming over the 0-2000 m interval than objective mapping methods for the period prior to 1990, but the overall warming rate is within the uncertainty of these and other studies. We also find strong evidence of the robustness of our results for the deep ocean by comparison to other estimates. Please see revised Figures 1, 2, and 5, and associated discussion. See also responses to detailed comments below.

Moreover, some of the outcomes have been linked to ocean processes and climate modes of variability without any further analysis steps.

We whole heartedly agree with this reviewer that an in-depth analysis is called for with regard to assessing climate modes, internal processes and changes to external radiative forcing in the context of the EEI over the 75-year period we consider here. We have adjusted the discussion to

emphasize the need to investigate these components further, however these are outside of the scope of the current study. While we do raise several points about these, they are merely raised to highlight the work of past studies that are relevant here. Please see also response to detailed comments below.

I therefore recommend rejection, and resubmission to another (more technical) journal for the new method after major revision.

The method is thoroughly explained and validated in the Methods section and Supplemental Material. We welcome any questions the reviewer might have on the methods.

Further comments below.

We thank the reviewer for the further comments which we address below.

L.11: The ocean is a dominant heat sink – could be also either energy – mostly in the form of heat ...

True. There is a lot of kinetic energy in the ocean as well, so we have clarified to refer specifically to the heat.

L.11-12: The Ocean sink is about 90%, so to be correct, add ‘recorded predominantly by’ ...

We have updated the manuscript accordingly.

L. 40: too generic message on ‘sparse measurements’, and does not recognizes the unique improvements of the observing system component with the implementation of Argo. Without this, science advances on the EEI could not have been made. This needs corresponding recognition in the revision of this sentence.

We agree that it is important to recognize the contributions of the Argo program here. We have revised this to say: “Over the past 15 years, the Argo program⁶ has deployed thousands of autonomous floats which provide continuous observations of the temperature in the upper half of the ocean, to a depth of 2000 m. This has allowed for a rapid convergence in estimates of OHC over the last fifteen years^{2,5} and increased confidence in calculations of the ongoing EEI in light of independent confirmation from modern satellite observations^{2,6,7}.”

L.76-84: These results contradict previous results, and Earth heat gain appears to be much lower than estimates elsewhere. Why is this the case? Why is this does not occur in the paper? There is no critical reflection on the outcome from this new product compared what has been published until now, and needs to be added.

In our updated analysis, our results generally fall within the range given by objective mapping studies in the upper 700 m (updated Fig 1b; note error shading is not given for other products but would be approximately as large as that for the ARANN). As for the 700-2000 m range, we mostly agree within uncertainty with two out of three of the objective mapping products (updated

Fig. 1c). In response to another reviewer, we also provided an additional figure here for the reviewers only to demonstrate that the dataset used in the reconstruction could have a significant impact in the 700-2000 m interval (Fig. R1). This figure is the same as Fig. 1c, but we have swapped our dataset of raw temperature casts from the WOD 2018 with one produced by L. Cheng et al. (2016) and used their vertical resolution of 42 depth levels instead of our 102 depth levels. Using their IGOT dataset, which contains far fewer observations in the 700-2000 m layer, leads to greater estimated warming. So, differences in the underlying datasets provides a possible explanation for much of the discrepancy in the 700-2000 m warming rates between IAP and our study, but a detailed intercomparison of all the methods is beyond the scope of the present study.

L.78-79: Fig. 1A shows an anomaly, and it is not straightforward to re-a-out the values for heat accumulation provided by the authors for both, the fluctuation, and the accumulation since 1990. Moreover, I am also puzzled by the strong-year-to-year variations at the end of the time series in ORAS4, which is not reproduced by the other time series (eg from 2010), what is happening there?

We have updated Fig. 1 to assist with reading, including adding more tick marks and changing units to ZJ.

ORAS4 transitions to being an “operational” model in 2010, likely leading to the strong year-to-year variations. Essentially, the model and data assimilation of the reanalysis is kept frozen and updates are provided via a different approach. Note however that in the revised manuscript, we have added many more products for comparison and as such we thought it best to remove the ORAS4 from comparison (see also response to comments below).

L. 85-87: The figures do not show a ‘broad agreement’ at all, and already the results presented in L.76-84 do not provide ‘broad agreement’. Or what is intended to say with such a very general statement on ‘broad agreement’? This is not clear. Moreover, a fundamental discussion is missing on the criteria for the choice of products used to compare the results from the new method. Why is IAP chosen? Why is ORAS4 chosen (by the way a more updated version ORAS5 already exists)? Particularly for the latter, this appears to be critical with respect to the statement the authors provide earlier in the manuscript (L.-47-49) – With respect to the information provided by the authors earlier in the text: why is now this reanalysis the one to be used as a critical reference for demonstrating a ‘broad agreement’? Since 2010 for example, i.e. when data sampling is high in the upper 2000m depth from Argo, ORAS4 shows much higher values than the other time series do. How is this explained? This discussion is much too vague.

Thank you for raising these points. In the revised manuscript, we include more focused analysis that compares our method to multiple objective mapping products, as these methods rely on direct in-filling of subsurface temperature data and have similar utility in climate studies as the present study. Our updated analysis demonstrates good agreement between our study and these objective mapping methods (updated Fig. 1). It falls within the range of estimates given by these

prior studies and gives similar warming rates to IAP for much of the record to within estimated uncertainty (see our next few responses for more detail).

The reviewer has a good point about our use of ORAS4 in the study, as it differs quite a bit from ORAS5 in its OHC results as well as from other available dynamical reanalyses. On consideration, we agree that the use of any particular dynamical model does not provide a compelling argument for broad agreement across various methods for estimating OHC. In our revised study, we do not put emphasis on the dynamical reanalyses, aside from retaining them in our updated Fig 5. Dynamical reanalyses and objective mapping methods fill different roles and should be judged on their own merits, as there are definite trade-offs across these methods that limit apples to apples comparisons.

While removing ORAS4 from the comparison, we have also added additional products that can capture changes in deep ocean heat content, including two passive transport methods (GF from Zanna et al., and OPT-0015 from Gebbie and Huybers), as well as estimates of warming rates from repeat hydrographic sections (see updated Figure 1 and 2).

In the end, we have removed the phrase "broad agreement" from the discussion here (except for the later period), and have instead included a more detailed discussion on the areas of agreement and disagreement with the previous methods. Please see revised Figures 1 and 2 and associated discussion. The "broad agreement" however does still apply when considering warming rates in different periods, as shown in Fig. 5.

L.88: Fig. 1B indicates that the new method dis-agrees strongly to what has been obtained in IAP, and almost does not show any warming prior to 1990 – that is what had been discussed earlier (L.76-84) and this statement appears to contradict and not consistent.

As mentioned above, we have revised our discussion to highlight the areas of similarity and difference between the ARANN and other methods (including IAP). So this seemingly contradictory statement no longer appears.

Nonetheless, we should note that the IAP and the ARANN do not disagree strongly. In the top 700 m our updated results are in broad agreement within the uncertainty ranges of objective mapping studies, including IAP. For instance, Cheng et al. 2017's Table S1 gives a warming rate of 0.09 ± 0.05 W/m² for 1960-1991 in the top 700 m and our study finds 0.06 ± 0.09 W/m².

It may also be useful to the reviewer to refer to Fig. R2 at the end of this document. This is a recreation of the SROCC Chapter 5 Table 5.1 with the results of the ARANN added for comparison in the style of Fig. 5 in our main text. When considering an ensemble of observations and modeled results, our estimate agrees within uncertainty.

L. 89: It is challenging to see an agreement for an 'oscillatory behavior' between ARANN and ORAS4? How do these time series correlate? Very vague statement for a result validation discussion.

Refer to comments above for more detail. Given the broad range of estimates of OHC given by various dynamical data assimilation models, including large differences between ORAS4 and ORAS5, we have opted not to include these as a major part of our updated analysis.

L.90-91: This statement appears to be not founded on what is shown in Fig. 1C – ARANN does not show any warming trend – at least what can be qualitatively derived from the figure (no trend values available to further found this discussion?) – whereas IAP does show a clear warming trend. Similar to my comment above this contradicts to what has been written in the paragraph above (L.76-84), and is inconsistent in itself.

Please see responses above, and revised Figures 1 and 2 and associated discussion. We are now clearly discussing the similarities and differences across a wide range of products in the revised manuscript, so this sentence has been revised (it no longer appears).

Again, we wish to note to the reviewer the uncertainties of the trends identified by prior studies and this study. Prior to 1960, IAP has very large uncertainty of roughly ± 70 ZJ below 700 m (Cheng et al. 2017 Fig. 4), so the trend over this period is not robust. For 1960-1991 Cheng et al. 2017's Table S1 gives a warming rate of the 700-2000 m layer of 0.02 ± 0.05 W/m², again no robust trend. Our updated results find a trend of -0.04 ± 0.05 W/m², which is in line with IAP. For 1991-2015, Cheng et al report a result of 0.23 ± 0.02 W/m², whereas we find 0.17 ± 0.06 W/m².

L.93-95: Results of Zanna et al. (2019, cited in this study) should be discussed here in this context, and relativize this statement accordingly (although shown in Fig. 1)...

We agree that this statement was potentially misleading since Zanna et al. (2019) also presented a data-based estimate of deep ocean heating. We have eliminated this statement.

...It would be good to understand for example whether this cooling had been reported elsewhere? Moreover, a discussion is missing on why the Zanna et al. do not show a similar signal?...

This is a good point. We have now added discussion that incorporates the work of Gebbie and Huybers 2019 and deep repeat hydrographic sampling from Desbruyeres et al. 2016. Likely the assumption of Zanna of ocean equilibrium in the 19th century does not allow for transient signals to fully propagate to the deep ocean, since this takes many centuries, unlike Gebbie and Huybers which uses a multi-millennia run to find deep Pacific cooling. Additionally, our comparisons to the deep hydrographic trends after 1985 look quite favorable (updated Fig 1 and 2).

...Moreover, as stated above, the comparison to an ocean reanalysis has limitations as stated by the authors in the introduction part. And this caution is further strengthened by looking at Fig. 2, in which the reanalysis result appears to strongly disagree?...

As mentioned above, we have removed the ORAS4 method from the comparison in the revised manuscript.

...Additionally, no trends are shown in Fig. 1, only the time series, although the change of trends over time are the major tool of argument for this qualitative comparison/discussion.

The discussion has changed significantly, and we think it is now easy to follow. Later on in Figure 5 we provide the trends.

L.95-96: Given that the layer above does not really show a warming signal, I am a bit puzzled about this statement. Has this been somehow quantified? This is not convincing from what is shown in Fig. 1 only. – particularly given that this is also highlighted as one of the major outcomes in the abstract (L.21)?

There is a small heat gain in the upper ocean that agrees with Cheng et al. within uncertainty. Deeper layers show a cooling trend over the same time period. We have supplied more detail in the revised manuscript.

L.105-107: Could the authors provide the analysis for the statement that the observed warming (or as shown the OHC anomaly) is consistent with vertical heat penetration from AMOC? This is not clear. Also not further reference is provide, nor a quantification to support this statement. Vertical penetration of heat is also not shown with Fig. 2.1

Thank you for pointing out that some of this discussion was ambiguous. We have revised our wording of this section to make the discussion more in line with the figures. Consulting Fig. 3d-f, it is apparent that the North Atlantic, Labrador, Greenland, and Norwegian Seas are experiencing more warming than the rest of the Atlantic basin at all depth intervals. Our revised discussion is more pointed and contextualized by prior studies.

L110-116: These statements based on Fig. 2 time series are an over-interpretation, and there is still major discussion on how or whether water mass formation with AMOC variability (eg Buckley and Marshall, 2016, doi:10.1002/2015RG000493). In addition, there is no study available which provides evidence on the use of a basin-wide OHC estimate as an indicator for AMOC variability and change. The link of upper ocean heat changes with AMOC variability has been discussed elsewhere (eg Zhang and Yang, 2017: <https://www.nature.com/articles/s41598-017-14158-6>), but specifically focused and analyzed to the processes. If a link to AMOC is thought to be established, a more precise analysis has to be performed. Likewise, there is still no consensus about the ‘warming hole’ and AMOC, and in my point of view this should be mirrored in the text, particularly when aiming to publish in a journal for a wider audience – see chapter 5 in IPCC SROCC for example on more content for this discussion (<https://www.ipcc.ch/srocc/>).

We have revised this section of our study as we do not wish to speculate beyond pointing out what is consistent with prior studies. We find patterns in agreement with the cited studies of our revised paragraph and include additional context about the possible role of internal variability in the region, although, as the reviewer states, there are still ongoing debates of the causal

mechanisms underlying observed patterns.

L.117-125: Recommend to include recent publication into this discussion: <https://www.nature.com/articles/s41467-020-15754-3>

Thank you for pointing out this study. We have added the recommended publication to our discussion in the revised manuscript to help contextualize and ground our findings.

L.122-124: see comment above – a more precise analysis is needed to connect the basin-wide OHC time series to such process discussions.

We agree. In the revised manuscript we refrain from establishing any causal links. We merely say: " The spatial distribution of warming in the deep Southern Ocean shows that the rapid warming over the past three decades is concentrated along the Antarctic margin, where Antarctic Bottom Waters form in the Weddell Sea, Ross Sea, and other marginal seas along the Antarctic shelf^{36,37} (Fig 3f)."

L.129-132: Also here, more analysis is needed for such as statement, and the link to climate variability indicators such as for example to the PDO. With this discussion, the link is not established. An example of how this could be done qualitatively is in Dieng et al., 2017, their figure 6 (DOI: 10.1002/joc.4996).

We agree that we need to be careful with our wording here. The revised manuscript does not make any causal arguments, but we merely wish to call attention to the work that has established the connections between climate modes and the heat content of the Indian and Pacific Ocean. The revised manuscript says: " Identifying the mechanisms responsible for these oscillations is beyond the scope of this study, but these could be related to changes in upper-ocean overturning circulation associated with the Interdecadal Pacific Oscillation (IPO)³⁹, which has recently been ascribed to changes in the strength of Pacific trade winds that affect eastern equatorial upwelling⁴⁰, as well as modifications to the winds in the tropical North Pacific⁴¹. These factors also appear to modulate heat transport into the Indian Ocean via the Indonesian Throughflow^{41,42}. " We agree that further investigation is merited, however, establishing a stronger causal link requires in-depth analysis using additional modeling products, which is outside the scope of the current study.

L.136-138: there is a recent publication linking to the paleo context and could support further discussions: <https://www.pnas.org/content/116/30/14881>. However, the cooling signals as shown in ARANN are not reflected in the other time series, any explanation?

Thank you for pointing out this study. Although it does not provide a direct point of comparison for our results, it provides a good example of how ocean circulation generates long-term climate modes that can influence the EEI, and we have added it to our discussion as follows: " Changes in the ocean overturning can also affect the EEI by modifying the rate of ocean heat uptake⁵⁴, which further leads to discrepancies between radiative forcing and EEI.

In the revised manuscript we have also incorporated the results of Gebbie and Huybers, as well as deep ocean repeat hydrography from Desbruyere et al, which we believe strengthens our findings and discussion of deep OHC change. The deep ocean cooling found by ARANN is also present in the multi-millennial reconstruction of Gebbie and Huybers (2019), while the more recent warming agrees with trends derived from repeat hydrographic surveys.

L.102-147 & Fig. 2: I am surprised to see no mention on the huge differences between the ARANN results and the other time series which have been claimed to be under ‘broad agreement’ in Fig. 1 – how can this be explained given that particularly the ocean reanalysis shows very different results – and which had been discussed as a further confirmation on ARANN deep ocean changes in the paragraph above? Why do these differences occur, and how can they be interpreted in the context of this discussion?

In the revised manuscript, we have added more comparison to other products in this section on the regional and basin-scale changes in OHC (Figure 2). We now consider multiple objective mapping methods for the 0-2000 m range, and we have revised the manuscript to include a thorough discussion of the similarities and differences across the various products, depth ranges, and time intervals. None of the methods produce identical results, which is not surprising since they use different methodologies. One of the key benefits of this study is that it provides an independent estimate using a method that is not similar to the standard objective mapping approach. We do not expect a priori that our method will produce identical results to previous objective mapping methods, nor would that necessarily be a desirable outcome. Nonetheless, using the IPCC SROCC report provided by this reviewer, we have assessed their Table 5, and found that we agree with the observational ensemble and the CMIP ensemble to within uncertainty for both the 0-700m and 700-2000m intervals and for every time interval in their table.

Below 2,000 m, we consider repeat hydrography and the passive transport method of Gebbie and Huybers to lend support to our findings in the deep ocean. As mentioned in our revised manuscript, reanalysis products are poorly constrained at depth and so we leave out the ORAS4 product in this revision.

L.151-153: I would be curious to see the results obtained for $d(\text{OHC})/dt$ using the other products for comparison. How different are they? Do they show a similar result? And additionally, a time series for net flux at TOA should be also included to comparison to further analyze the robustness of the results. The CERES product for the most recent period, and a long reconstruction such as for example from Allan et al. 2014, <https://agupubs.onlinelibrary.wiley.com/doi/full/10.1002/2014GL060962>. In addition – this is what would be required to ‘close the budget of the Earth energy imbalance’ – this is what the title promises – but the entire paper does not provide any analysis step of discussion linking to ‘budget closure’.

We agree that it is a good idea to compare our dOHC/dt estimate to the TOA radiative flux estimates. We now incorporate the reconstruction of Allan et al. (which is based on CERES in the later period) into Fig. 4. We have included additional discussion of this comparison as follows: "The timing of this shift, the magnitude of the implied EEI, and the temporal variability of the EEI agree well with quasi-independent estimates of the top of atmosphere (TOA) net radiative flux⁶ for the period 1985-2016. The only time periods where the TOA net radiative flux lies outside the 2σ uncertainty of the ARANN-estimated EEI are in the early 1990s just after the Pinatubo eruption, when the TOA radiative flux is more negative than the dOHC/dt, and during the early 2000s when the dOHC/dt shows an upward jump that is opposed to a drop in the TOA net radiative flux (Fig. 4)."

We have also added additional objective mapping products to compare OHC for Fig 1 and 2, although these are not added to Fig. 4 to avoid cluttering the plot, and because they do not include the deep ocean below 2000 m and so do not close the ocean heat budget.

L.161-164: Also as discussed above, there is further evaluation needed: why not adding the climate mode indicators to the figure to further underline the discussion? Neg./pos. EEI values are not news knowledge, but a more in-depth attribution study would be of strong value for climate science.

An attribution analysis would be a study in and of itself, and thus lies outside the scope of the current study. We provide an estimate of the EEI for the period 1946-2019, which is a novel contribution, as it is derived from a full-depth OHC estimate that includes the deep ocean. Other studies have considered shorter time periods, been modeled results, or covered the 0-700 m OHC. While an attribution study would be most welcome, we consider that our finding of deep ocean cooling reducing the EEI for the 20th century must be prioritized in the present study.

Also, the evolution for ‘El Chichon’ shows an increase towards 0 EEI. How can this be interpreted?

We have added some discussion of the mismatch to the timing of El Chichon in the revised manuscript. It would be interesting to investigate and there was some minor volcanic forcing just prior to 1982, but ocean internal variability may explain this discrepancy as well. Likewise, the temporal resolution of the ARANN-estimated EEI is on the order of 1 year, so we can't expect an exact match to the timing, although there should be an approximate match within ~1 year or so, as shown in Fig. 4.

L172-176: This is not the case, see discussions above, and the authors own statements. Moreover, such a discussion would only be valid if the other results would have been added into Fig. 4 as well.

These lines are in reference to Fig 5, and the statement was accurate (although it has been revised in the updated manuscript). Trends in the warming rates are commonly reported in other studies and are incorporated into Fig. 5, for which we find reasonable agreement across methods. Our estimate for 1960-1990 and 1960-2019 is on the lower end, partly due to deep ocean cooling, but does not represent an overturning of the consensus. The revised manuscript now states:

"Objective mapping, passive transport methods, and climate models have generally estimated small warming rates over the 1960-1990 period, but mostly agree within the ARANN estimates within their respective 2σ uncertainties (Fig. 5)." And we also state "Overall, the ARANN results support a broad consensus across almost all products of accelerated warming over time (Fig. 5), but they also suggest that previous estimates of ocean warming may have been biased too high prior to 1990, in part due to the neglect of deep ocean cooling during this time period. Accounting for this deep ocean cooling leads to a delayed onset of positive EEI until roughly 1990."

L.180-182: Yes, all other products / method disagrees with early ARANN estimates. Why is this the case?

Again, they do not disagree. Please see revised Fig. 5, as well as Fig. R2 at the end of this document. The issue is more subtle: The ARANN estimate lies at the lower end (within uncertainty) of the other estimates, and suggest a delayed onset of positive EEI. This is what we are showing here, and we are not overturning a consensus on ocean warming rates prior to 1990 (which would require disagreement beyond the 2σ level, and would also require some community consensus on the deep ocean OHC which has not been incorporated in other studies).

In general, different products and methods do disagree somewhat with each other, due to the different methods and datasets used, which is an indication of the inherent uncertainty in estimates of OHC. Determining the reasons for these differences would require a rigorous intercomparison of the methods, including access to raw datasets and source code for these various studies, which is a topic for a separate study.

If it is due to the fact that ARANN provides an improved estimate for deep ocean changes, why is this not further validated, either through comparison to regional (if available) deep ocean measurements,...

These deep ocean measurements have already been assimilated into the ARANN (see Methods). We have validated that we can reconstruct deep ocean trends from the observed sparse data distribution using results from climate models (see Supplementary Materials). We also show in the revised manuscript that the trends we derive in the deep ocean are consistent with those derived from repeat hydrographic measurements (see revised Fig. 1 and 2), and in agreement with the passive transport method of Gebbie and Huybers (2019) prior to 1990 in the Pacific and Indian Oceans.

...or under a physical budget approach (e.g. by comparing with net flux at TOA ?), or the sea level budget while using reconstructions, etc.?

The revised manuscript includes a comparison of the TOA net flux from 1985-2016 in Fig. 4.

A more robust validation of the results are needed to assure robustness of the outcomes.

As before, we believe that the reviewer is referring to a comparison with previous studies, rather than validation (the Supplemental Material provides in-depth validation of our method by applying it to temperature fields from ocean models, showing the ability to recreate modeled

temperature fields from the observed data sparsity on global to basin scales at all depth intervals, even in the presence of realistic geophysical noise).

The revised manuscript now provides much more comparison with previous studies, including additional objective mapping products, passive transport products, and repeat hydrography methods (Fig. 1 and 2), TOA net flux products (Fig. 4), and essentially all the available estimates for ocean warming rates from the literature in Fig. 5.

L.189-192: This statement needs more clarification, and it is not clear on how the 3 volcanic eruptions induce a ‘lack of ocean warming’ – what does this mean? And how can a climate forcing that acts on short (1-2 years max) counteract a signal over the period 1950-1990, thus several decades? Please further clarify.

We did not claim that volcanic eruptions induce a lack of ocean warming, but that they could help contribute to it. Additionally, estimates of instantaneous forcing terms in CMIP5 simulations account for many more than 3 volcanic eruptions. The 3 mentioned are merely the largest, but others play a role as well. Different models for volcanic aerosols have different effective times, so this isn't necessarily an impact over 1-2 years, especially in the presence of multiple concurrent volcanic events in the 20th century.

Overall, volcanic aerosols lower the average forcing during the latter half of the 20th century, which would be reflected in less ocean warming, producing a “counteracting effect” to anthropogenic forcing. This would partly explain what we find in our study, as the rapid increase in ocean warming coincides well with a quiescence of volcanic activity at the end of the 20th century. However, volcanic aerosols alone do not fully explain the “lack of ocean warming”, which is why we propose the influence of longer term modes of climate variability or an underestimation of anthropogenic aerosols to account for the difference.

The relevant part of the revised manuscript that mentions volcanic forcing reads as follows: "On the other hand, the recent accelerated warming since 1990, and in particular since year 2000, is consistent with the dominant effects of anthropogenic greenhouse gas forcing and negligible volcanic aerosol forcing^{1,55} during this time period."

L.200-203: This statement needs also further explanation to understand the main message, this is very general, and not clear.

Here, we were referring to our reconstruction of the sub-decadal variability of EEI that is not consistent with anthropogenic forcing or major volcanic events. As this point was out of place here, we have moved this earlier in the discussion where we are discussing Figure 4 (which now also shows a comparison to the net TOA flux of Allan et al. which shows similar levels of variability).

L.231: Not clear on the use of the climatology: Using only a most recent climatology, which is comparable warm (ie on the top of the strong warming observed during the past decade)

could potentially explain why a low (or no trend) is observed at the beginning of the time series? How robust is the method compared to different climatologies (eg. starting with a ‘cool climatology’ for 1960-1970? And one 1970-1990, and then one 1960-2018)? Would this effect the global trends of ARANN?

This is an excellent question, also raised by other reviewers. To account for the effects of climatology on the inferred OHC changes, we have updated our analysis to utilize 6 decadal climatologies [1955-1964 1965-1974 1975-1984 1985-1994 1995-2004 2005-2017] from the World Ocean Atlas. In addition, at the suggestion of another reviewer, we have addressed concerns about our 6-month time window for binning by creating an equal number of realizations that bin Jan-Jun and Jul-Dec as well as Apr-Sep and Oct-Mar. Over 240 ensemble members, we do not find that choice of climatology has much of a significant impact on global trends in ARANN (see updated Fig. S14).

Reviewer #3

This study constructs a state-of-the-art estimate of historical ocean warming between 1946-2018. This reconstruction is unique in that it covers the entire water column (full depth), and thus includes the relatively large volume of the global ocean that is generally unaccounted for in estimates of global energy imbalance. The authors do so by applying novel machine learning techniques to populate regions of sparse data coverage, which is a particularly significant accomplishment in the deep and abyssal ocean where data coverage is quite limited is notoriously difficult to constrain. The application of this methodology to ocean heat content estimates is new and performs quite well. The authors find that the addition of estimated abyssal heat uptake implies very little total imbalance in TOA radiation between 1950-1990, which conflicts with other (less constrained) estimates, a result that will interest a broad range of climate and ocean scientists. I really enjoyed reading this paper and think it would be appropriate for Nature Communications, both for its relevance to a number of fields as well as the implications of the findings.

Thank you for your feedback and for the important points raised below.

I have one technical issue and one general point the interpretation of results, as well as several minor questions/comments I'll list below.

Please see our detailed responses below. In short, we have produced an updated analysis that considers 6 different decadal climatologies to allow us to better quantify uncertainty in the ARANN estimate. Additionally, multiple objective mapping products as well as deep ocean datasets are considered as points of comparison in our updated study. These additions should bolster our conclusions about the nature of the EEI over recent decades, including impacts from changes in the deep/ abyssal ocean.

Technical issue:

Lines 231-253: I am having trouble understanding why the choice of seasonal climatology, and specifically the use of a climatology from the latter part of the record (2005-2017) doesn't bias the result. I see that you do explore the sensitivity of results to the choice of climatology, concluding that it doesn't matter much, but I am confused about how this could be. If I'm understanding correctly, ocean warming as presented in the paper (i.e., Figs. 1-5), is defined as the anomaly between the data and this seasonal composite. If so, aren't you subtracting a climatology that has a higher average temperature, given the fact that it was calculated using data in which the ocean has warmed substantially from earlier in the record (at least in some regions). I list this concern first because the negative heat content anomaly earlier in the record (1950-1990), particularly in the deep ocean, is an important result of the paper; I am having trouble understanding how this isn't a unavoidable artifact of your methods.

Thank you for raising this point. Originally, we considered two climatologies. One for all years <1946-2017 and the other for just 2005-2017. However, the <1946-2017 climatology would be overly weighted towards the 2005-2017 time period due to better sampling coverage afforded by the Argo array during this time, and so differences between these two climatologies may have been muted.

In our updated analysis, we use 6 different decadal climatologies [1955-1964 1965-1974 1975-1984 1985-1994 1995-2004 2005-2017] from the World Ocean Atlas. This ensures that our results are not biased by using a climatology from any particular time period. At the suggestion of another reviewer, we also used an equal number of realizations with a six-month binning that average Jan-Jun and Jul-Dec as well as Apr-Sep and Oct-Mar. These choices are all intermixed in our ensemble of simulations. In the end, this leads to slightly larger error bars in our analysis, but the major findings remain unchanged.

Updated Fig S14 compares the two extremes of the climatologies. Using the 1955-1964 climatology produces less warming in the top 2000 m than the 2005-2017 climatology. However, the 1955-1964 climatology produces slightly more warming below 2000 m, partly compensating for lower warming rates in the shallower depth intervals. Overall, there is only a small difference of ~40-50 ZJ heat gain for 0-5500m OHC from 1946-2019. Notably, the trends in terms of the shift from negligible warming to rapid warming are the same regardless of climatology.

I am likely missing something, but I would be more convinced if you were to provide a more physical explanation for the insensitivity to the seasonal climatology. Further, I wonder if the equivalent to Fig. S14 could be performed with one the models used in the study, given the issues with the observational data coverage from the earlier climatology that Fig. S14 is based upon. However, if there is a simple explanation for the insensitivity to the use of climatology, further analysis isn't necessary. Regardless, a clear statement about how temperature anomalies were defined should be in the main manuscript (somewhere between L39-55).

There is some sensitivity to climatology that, hopefully, has been better clarified by our updated Fig. S14. We find that using a more recent climatology produces more total warming, consistent with the findings of Cheng and Zhu (2015). Our revised results fully propagate uncertainty in the climatological choice to our OHC estimates.

A simple explanation for the relative insensitivity is that the ARANN can recreate small temperature anomalies equally well as large temperature anomalies, and so it doesn't matter too much how the temperature anomalies are defined, as long as it is done so consistently across the time period in order to have a common baseline of OHC for that time period.

We have included a statement at the end of the revised introduction to point out that the choice of climatology is included in our uncertainty estimates: "...and choice of reference climatology used to define the temperature anomalies^{25,26} (Supplementary Fig. 14)." The reader will find details on how the anomalies were defined in the Methods section: "After binning the temperature data to the WOA grid, we subtracted a monthly temperature climatology to create a field of monthly

temperature anomalies (Supplementary Fig. 1a, Step 4). For this step, we used one of six WOA decadal climatologies covering years [1955-64 1965-74 1985-94 1995-2004 2005-2017]. These climatologies are monthly in the top 1500 m and seasonal below that. The choice of climatology has a relatively small impact on the global OHC estimate, which becomes more significant in the deep ocean and further back in time (see Supplementary Fig. 14)."

Interpretation issues:

I found a few areas where the paper would be enriched by clearer interpretation of the results.

Most generally, and given the emphasis of the paper, I feel the work could be strengthened by a more concrete comparison between the EEI presented here to estimates of radiative forcing. Specifically, the results in Fig. 5 and discussed in the manuscript imply that previous estimates of energy imbalance in the climate system over much of the historical record have been biased high. The authors discuss in (lines 184-206) that this finding is at odds with estimates of positive radiative forcing. It would be interesting to see these radiative forcing estimates on Fig. 5, giving a more quantitative understanding of how significantly variability can influence the energy budget.

This was perhaps a misleading statement as it may have inadvertently implied that the EEI and the radiative forcing should be the same. This is not the case, and we have clarified our discussion as detailed below. Radiative forcing estimates are generally calculated relative to the pre-industrial era and the impact of this forcing is not immediately imprinted on the EEI because the Earth system is not in equilibrium. Due to the timescales of overturning in the ocean, propagating the entire forced climate signal from the surface to the interior may require decades or more, especially in the case of consistent anthropogenic forcing from aerosols and greenhouse gases. Comparing the radiative forcing estimate apples to apples with the EEI may therefore be inadvisable on short timescales since some lag between forcing and response would be expected.

The revised manuscript clarifies this discussion as follows: "Additionally, given that anthropogenic radiative forcing has remained positive and continued to grow in magnitude over the past century¹, the lack of global ocean warming over the period from 1950-1990 may seem counterintuitive at first, but the Earth's climate system is not currently at equilibrium. Due to the timescales of overturning in the ocean, propagating the entire forced climate signal from the surface to the interior may require decades to centuries to manifest as signals in the deep OHC^{8,10}, implying that the EEI is also modulated by changes in external forcing on multi-decadal time-scales." See also our further discussion in the last paragraph of the conclusions.

In all, the findings of this study should motivate a deeper investigation of Earth's climate sensitivity, which measures the strength of the surface warming response versus the difference between the radiative forcing and EEI. However, this topic deserves its own in-depth analysis that is beyond the scope of this paper.

Also, are the cooling estimates here at all consistent with Gebbie & Hubers (GRL, 2019)? It would be good to include discussion of the relationship of deep temperature anomalies to historical surface temperature anomalies in ventilation regions, in addition to the discussion of variability.

We have included in our revised manuscript comparison to the estimates of Gebbie and Huybers in the deep ocean and find that these are largely consistent for the Pacific and Indian Oceans. These are regions where past surface cooling would potentially be preserved due to the long overturning time-scales. In the Atlantic and Southern Oceans, where deep water formation occurs and vertical mixing leads to water masses of different ages being recombined, we do not find nearly as good agreement with Gebbie and Huybers. The circulation in these basins is well-documented to have undergone significant changes, such as changes in the Atlantic overturning circulation, and contraction of Antarctic bottom waters. We have updated our discussion to include these considerations (see discussion associated mainly with Figures 1 and 2).

Likes 102-116: I found myself re-reading this passage multiple times to understand your points. I think it is confusing because you have partitioned the data into two epochs in time, and several layers in depth, and skip back and forth between them in this discussion. It would help if you were more consistent in summarizing results around the two epochs in time you use throughout the paper. I'm also confused with your interpretation: you attribute the deep Atlantic transition from cool to warm anomalies to a possible slowing of the AMOC after 2005, but also attribute the dipole in upper ocean temperatures prior to 1990 to a cessation of deep-water formation. Are you not attributing two different features to the same phenomenon, but at different and inconsistent times?

We agree that the original discussion was not structured in a clear manner. The revised manuscript presents this in a clearer manner. In particular, we have labeled the panels of the figures so that we can refer to the exact time/depth period without causing confusion for the reader. The revised discussion is as follows: " Fingerprints of ocean circulation changes are also apparent in the spatial distribution of warming rates (Fig. 3). Prior to 1990, the upper 700 m of the subpolar and polar North Atlantic was cooling, while the Gulf Stream extension region was warming (Fig. 3a), consistent with trends that have previously been identified as fingerprints of reduced deep convection in high-latitude deep water formation regions^{27,28}. Since 1990 there has been coherent strong warming throughout most of the Atlantic basin, except for a small patch of cooling in the center of the North Atlantic subpolar gyre (Fig. 3d). Warming of the North Atlantic from 1990-2005 has previously been linked to a surge in the Atlantic Meridional Overturning Circulation (AMOC) after 1990 (²⁹), although this has been followed by a decline in the AMOC after 2005 and cooling of the subpolar gyre³¹, potentially contributing to the cooling trend identified in the central subpolar gyre in Fig. 3d. After 1990 there is also pronounced warming at mid depths (700-2000 m) throughout most of the Atlantic, concentrated more strongly in the subpolar north and south Atlantic (Fig. 3e). This is consistent with the mean overturning circulation transporting surface warming to intermediate waters, since these regions are close to the formation regions for North Atlantic Deep Water in the North Atlantic²⁹ and Antarctic Intermediate Waters in the South Atlantic³⁰."

The same comment, regarding being more clear/consistent about the time periods compared, applies to the following few paragraphs on regional results (117-147).

We have revised our discussions accordingly. In particular, reference to particular sub-panels of the figures should help to clarify exactly what times and depth periods we are referring to.

Lines 159-162: Have you compared your results to an ENSO or IPO index? If a relationship existed there, it would be a nice addition.

Thank you for this suggestion. We agree that this is certainly an avenue worth pursuing as a future direction, but such an attribution study is outside the scope of the current work. Establishing a link between climate modes and natural variability in the EEI is a highly desirable outcome, but is not as straightforward when one considers the time-lag in propagating climate signals from the surface into the ocean interior and regional differences in this delay.

Lines 200-204: Is it possible to place the magnitude of variability in heat content found here to other estimates? Even, perhaps, the models explored the study (despite their flaws in deep ocean physics and water mass properties). Or historical records of deep ocean temperatures? I am left wondering how the findings here compare to other estimates, though I realize one of the points of the paper is that similar estimates are lacking from the literature. However, even a qualitative comparison of your results to others would be

We have revised our manuscript to include an expanded number of datasets to compare to our estimate, along with additional discussion. These additional datasets should solidify some of our original points, especially in the case of the deep ocean.

Small comments:

Line 117: Define what you're calling the Southern Ocean (in fact, the same applies to Atlantic and Indo-Pacific). I assume all regions are partitioned at 30S?

We now specify this in the caption to Figure 2. We use the WOA 1 degree mask to demarcate the basins. The Southern Ocean is partitioned at 50 S.

Line 163: This sentence is a bit misleading. Variability is still important later in the record, it just doesn't override forcing. Please reword.

We have updated the manuscript accordingly. The relevant statement regarding the EEI variability is now: " This supports prior studies that point to internal modes of climate variability such as ENSO^{2,7} and the IPO^{4,39}, or natural variability in solar irradiance⁴⁶ as factors influencing sub-decadal changes in the EEI for this 75-year record."

Line 165: You discuss this as though the consistency is a confirmation of the methods, but

don't Fig. 1 and Figs. 4-5 have to be consistent by construction (aren't you just taking the derivative of OHC from the same dataset)?

We did not mean to imply this as a confirmation of the methods. To avoid confusion we have removed this statement. As a note, fitting a trend to the OHC and taking an average of the $dOHC/dt$ will not necessarily yield the same result, but for long enough times should be similar.

Line 177 and Figure 5: Please note in the caption that the time periods averaged over in Figure 5 are different from the previous figures. I was confused for a bit about how Fig. 4 and Fig. 5 weren't at odds.

We have revised the time periods include in Fig. 5 to be more consistent with our discussion, in particular by breaking out the 1960-1990 interval.

Line 199: add "the" year 2000.

Revised.

Line 257: what do you mean by the characteristic length scale in this context? Of the sinusoids?

That is correct. We have updated this for clarity: " Due to how these sinusoids are constructed, they each represent a basis function that is merely a 2-dimensional array of numbers between -1 and 1 with a characteristic length scale ranging from roughly 1,000 km to 18,000 km."

Line 265: here you refer to the horizontal spatial variability, correct? If so, add the word "horizontal" or "lateral" or something.

Correct. We have added "horizontal".

Line 272:274: This is an interesting point. Does this provide an opportunity for physics-based discovery, specifically, as an independent diagnostic of implied timescales of ocean memory?

There are certainly ways to obtain this kind of diagnostic using the framework we have proposed. For instance, within the model, weights (free-parameters) that are associated with time-steps that have stronger correlation with the current time step will be larger, though this is somewhat qualitative. This can be applied regionally (for instance the North Atlantic) by subsampling the temperature observations to produce modeled weights that indicate different time-scale dependencies.

Related to this point, I have some confusion around lines 283-291. I'm somewhat confused about the iterative incorporation of additional vertical layers. Does it not matter that characteristic scales of vertical gradients in, say temperature, will differ between shallow and deep regions of the ocean? Should we expect an algorithm training on shallower data

to be applicable deeper in the ocean? Could you please expand here on why you incorporate progressively deep data in this way?

We have expanded on these points in our discussion of the methods of our revised manuscript by including: “Vertical mixing is important in certain regions and contributes to deep water formation, so surface warming of the ocean would be expected to display some imprint on the layers below. Because the ARANN is iterative, it optimizes for each depth interval during its sweep from the surface to seafloor. Relationships identified by the ARANN at depth will therefore evolve from those found at the surface.”

Fig. S2A Caption: “near the.”

Revised

Fig. R1. Warming rates from SROCC Chap. 5 versus ARANN

Warming rates for 0-700 m and 700-2000 m from the ARANN method, compared to those from mapping products and CMIP5 models as stated in the IPCC SROCC Table 5.1.

Fig. R2. Impact of dataset choice on estimated OHC

Blue is using the 2018 World Ocean Database as described in the manuscript (same as main text figures). Red is using the IGOT database that was used by Cheng et al. (2016) for the IAP mapping product. And black is the IAP result.

REVIEWER COMMENTS

Reviewer #1 (Remarks to the Author):

The manuscript, "Closure of Ocean Heat Budget Reveals Delayed Acceleration of Earth's Energy Imbalance," by Bagnell and DeVries, has been revised with figures that give more context into deep ocean temperature trends, with improvements to the written presentation, and with statistical analyses that go beyond the typical high-profile paper. The improvements are interesting and valuable, and it is worth re-evaluating the principal claims of the manuscript and whether Nature Communications is the right venue for publication.

1. The title of the manuscript highlights the "delayed acceleration" of Earth's energy imbalance. In this manuscript, nearly all of the global ocean heat uptake since 1946 has occurred after 1990, in contrast to existing heat uptake estimates that had warming before 1990. In the response to reviewers, the authors note that they "don't think that the results presented here represent an overturning of the community consensus." The issue is that the uncertainty level on pre-1990 ocean heat content is too high due to observational sparsity and biases to statistically distinguish their ARANN analysis from previous analyses. Thus, the null hypothesis that all ocean heat content estimates prior to the satellite era are actually consistent with each other still reigns. One reconstruction shows delayed acceleration of Earth's energy imbalance; many more do not. It is my opinion that the authors bear the burden of explaining why the delayed acceleration of their title is to be believed over the multitude of other estimates that do not show such a delay. The authors respond that this is "certainly beyond the scope" of this work. I suggest the contrary: that the robustness of the main finding and justification for the choice of title is central to a high-profile manuscript. Is delayed acceleration revealed or is it a statistical artifact?

2. The manuscript goes beyond previous reconstructions by taking into account deep ocean heat content trends with "interpolated subsurface temperature data." (The meaning of "interpolated" here is ambiguous.) They find a decrease in ocean heat content below 2 km depth, especially in the Indian and Pacific Oceans, which is a novel result (and probably the most important story of this manuscript). Deep ocean temperature trends are linked to the delayed acceleration of the total ocean heat uptake. Figure 1 contradicts the statements in the text, because the 30 ZJ drop in deep ocean heat uptake from 1950-1990 (Figure 1d) is a small value relative to the community-consensus 100 ZJ increase in upper ocean heat uptake over the same time period (Figure 1b). The 30 ZJ decrease doesn't compensate for the upper ocean heat content increase. Instead, the delayed acceleration in ARANN is primarily caused by the lack of increase in upper ocean (not the decrease in deep ocean) heat content. Furthermore, Supplementary Figure 14 indicates that shifts in ocean heat content by 50 ZJ can be caused by changes in the ocean climatology used by ARANN. These shifts are greater than the 30 ZJ decrease in deep ocean heat content that are central to main finding #2. Thus, these deep ocean temperature trends are not robustly indicated as the dominant cause of delayed acceleration.

Summary remarks:

High-profile papers have the burden of being simultaneously novel, important, and statistically rigorous. Many such papers handle these constraints by doing minimal statistical analyses at the cost of weakening their impact. The authors are to be commended for completing a more quantitative statistical treatment; however, the presentation of the findings is not consistent with the quantitative results. I evaluate the deep ocean analysis to be interesting and valuable, but I cannot recommend publication in the current form.

Minor issues:

Figure 2: the chosen baseline of the curves, which is ambiguous, will significantly affect the visual impression of the figure.

The Little Ice Age is most commonly defined to be a several hundred year interval.

Reviewer #2 (Remarks to the Author):

The paper 'Closure of Ocean Heat Budget Reveals Delayed Acceleration of Earth's Energy Imbalance' has significantly improved after major revisions, and I recommend publication after further minor revision, see comments below.

Further Comments:

Title – and general comment as mention also below: The use of the wording 'budget' is confusing, and misleading. This study does not provide a budget analysis, it provides an evaluation of global ocean heat content. I strongly recommend to refine this wording as I suspect large confusion within the community when used in its current state. If the authors insist to keep the wording, a clear definition for what 'budget' stands for here, and why and how this study addresses a 'budget closure' analysis would need to be added for clarification.

Other general comment (conclusion part): I strongly recommend to further discuss these new results in reflectance with previous estimates, and to particularly provide further links / ideas for needed research paths forward, and improvements and further evaluations needed.

P1, L11-12: There is a principal mis-concept in this statement. The EEI at TOA is currently measured by remote sensing. There is a challenge to obtain the absolute value with the CERE program (anchored by ocean heat content), but it is the most precise measure today on the variations of the EEI. Then, the heat inventory to obtain the heat stored in the Earth system form a positive EEI is derived from the Earth heat inventory, and as stated, the majority is stored in the ocean. Revision is needed accordingly.

P1, L14-15: Which is the percentage today estimated from the deep ocean contribution? More than the 10% which are missed out by considering the ocean only in this concept for the Earth heat inventory, particularly when also accounting for the uncertainties?

P2, L45-48: This statement is not linked to any reference, and also it is not clear on how the energy budget closure is approached here: Is it the close correspondence between net flux at TOA and the Earth heat inventory? If yes, I am puzzled because I do not see recent publications pointing to a new 'missing energy concept' (DOI: 10.1126/science.1187272) at timescales the authors are aiming to resolve (long-term) – where is this statement – and the major motivation of the study (incl. the title) based on? If the EEI budget constraint is not addressed here, the concept on 'energy budget' in this study needs to be clarified.

P2, L47-48: this does not leave the energy budget open, but deep ocean contributions to global ocean energy storage remains uncertain – this would be a correct way to phrase this, see comment above.

P4, L133: Would recommend to add the uncertainty range to this estimate of 30-40 ZJ – general recommendation for the entire document, will not raise every time.

P7, L295 recommend to use wording 'quasi-equilibrium' instead

P8, L320-321: recommend to add also the recent paper of Kramer et al. into this discussion (<https://doi.org/10.1029/2020GL091585>)

Reviewer #3 (Remarks to the Author):

I appreciate the author's further sensitivity analysis to the choice of climatology, and incorporation of different climatologies into the OHC estimations presented in the study. I also thought the revised passages of the manuscript were much clearer and easier to place in the context of previous estimates. I am generally satisfied with the revision and still feel the paper is appropriate for the Nature readership. I have one remaining issue with the paper, and that is, despite substantial analysis of error, there is relatively little discussion or disclosure of it in the text. Simply adding error ranges into estimates, and a few disclaimers, would strengthen the reader's understanding of where the methods are more or less certain.

Specific points:

L99, L101, L133, L144 added error estimates (+/-).

L301: "so the lack of global ocean warming..." add something along the lines of "implied by our new methodology."

L333: I'm not sure I'd say there's relatively little impact- earlier in the record, the average implied OHC anomaly is ~50ZJ different between climatologies, basically the same as the total OHC anomaly at during that time in Fig. 1. Please amend to something more clear- the climatology has a small impact over the final 3 decades and a more significant impact, contributing to the greater uncertainty in the OHC estimate, earlier in the record and deeper in the ocean.

Reviewer #1 (Remarks to the Author):

The manuscript, "Closure of Ocean Heat Budget Reveals Delayed Acceleration of Earth's Energy Imbalance," by Bagnell and DeVries, has been revised with figures that give more context into deep ocean temperature trends, with improvements to the written presentation, and with statistical analyses that go beyond the typical high-profile paper. The improvements are interesting and valuable, and it is worth re-evaluating the principal claims of the manuscript and whether Nature Communications is the right venue for publication.

1. The title of the manuscript highlights the "delayed acceleration" of Earth's energy imbalance. In this manuscript, nearly all of the global ocean heat uptake since 1946 has occurred after 1990, in contrast to existing heat uptake estimates that had warming before 1990. In the response to reviewers, the authors note that they "don't think that the results presented here represent an overturning of the community consensus."

We stated this response because we don't believe that any single study could either establish a consensus or overturn a consensus, regardless of what level of uncertainty was reported by that study. The consensus must be built from multiple, independent methods arriving at a similar result. Thus, the OHC estimates from the present study, along with those of previous studies, must be properly weighed and considered when establishing a consensus. Likewise, future studies will serve to further refine such a consensus.

As the reviewer correctly mentions, large uncertainties remain in the exact trajectory of OHC over the 20th century. Our study provides one plausible trajectory that in one sense compliments the broad consensus established by prior research, as it shows accelerated warming after 1990. In another sense, it challenges that consensus because our study indicates a stronger rate of acceleration in the EEI than prior studies have found, mainly due to estimating less warming prior to 1990 than other studies. We hope that these findings, when properly contextualized with their uncertainties (see below), may serve as a catalyst to inspire further investigation into the trajectory of OHC over the 20th century.

The issue is that the uncertainty level on pre-1990 ocean heat content is too high due to observational sparsity and biases to statistically distinguish their ARANN analysis from previous analyses. Thus, the null hypothesis that all ocean heat content estimates prior to the satellite era are actually consistent with each other still reigns.

The reviewer raises an excellent point that the uncertainty levels in the pre-1990 ocean heat content are too large to statistically distinguish our estimate of no warming rate from previous estimates that showed small but positive warming rates. We agree that it is important to explicitly point this out to the reader.

We therefore explicitly highlight this uncertainty in several places throughout the revised manuscript. In the discussion we now state (emphasis added): "The ARANN reconstruction of

full-depth OHC provides an internally consistent framework for monitoring EEI over time, showing that the Earth energy budget was in quasi-equilibrium, with substantial decadal variability, for the four decades from 1950-1990. *The warming rate from the ARANN does not differ from that derived by objective mapping methods with statistical significance, and previous studies already support a slower ocean warming rate for the 1950-1990 period relative to the 21st century (Figure 5).* However, due to the combination of a smaller estimated change in 0-2000 m OHC for 1950-1990 and the contribution of deep ocean cooling, the ARANN implies a stronger and later shift toward accelerated EEI than previously recognized, and raises the question as to what may have caused this climate shift."

And we also note the large uncertainties directly in the abstract (emphasis added): "These results suggest a delayed onset of a positive Earth energy imbalance relative to previous estimates, *although large uncertainties remain.*"

One reconstruction shows delayed acceleration of Earth's energy imbalance; many more do not. It is my opinion that the authors bear the burden of explaining why the delayed acceleration of their title is to be believed over the multitude of other estimates that do not show such a delay. The authors respond that this is "certainly beyond the scope" of this work.

We do not necessarily think that our estimate of the OHC trajectory should be believed over that determined from previous studies. Nor do we think that previous studies should necessarily be believed over our results. What we do believe is that our method provides one plausible trajectory for OHC changes over the past 75 years, based on a fully validated and consistent method, and that our estimate suggests several novel findings that are of interest to the climate community, including deep ocean cooling and lower-than-previously-thought upper ocean warming prior to 1990, and that large uncertainties in the precise OHC trajectory remain.

We also believe that these findings may inspire future work that will critically examine the methods and datasets used across a wide variety of OHC reconstructions, in order to more precisely determine the reasons for the differences across methods and their relative performance on synthetic and real-world datasets. It is these types of studies that would be needed to determine the relative believability of any particular OHC reconstruction and that are beyond the scope of the current study.

While it is beyond the scope of this study to resolve these particular issues, it is also our responsibility to ensure that we clearly state the current limitations and how they impact the interpretability of our findings here. Thus, we have strived to clearly state the uncertainties in our revised manuscript and call attention to them explicitly where applicable.

2. The manuscript goes beyond previous reconstructions by taking into account deep ocean heat content trends with "interpolated subsurface temperature data." (The meaning of "interpolated" here is ambiguous.) They find a decrease in ocean heat content below 2 km depth, especially in the Indian and Pacific Oceans, which is a novel result (and probably the

most important story of this manuscript). Deep ocean temperature trends are linked to the delayed acceleration of the total ocean heat uptake.

Figure 1 contradicts the statements in the text, because the 30 ZJ drop in deep ocean heat uptake from 1950-1990 (Figure 1d) is a small value relative to the community-consensus 100 ZJ increase in upper ocean heat uptake over the same time period (Figure 1b). The 30 ZJ decrease doesn't compensate for the upper ocean heat content increase. Instead, the delayed acceleration in ARANN is primarily caused by the lack of increase in upper ocean (not the decrease in deep ocean) heat content.

Yes, our study does find less warming from 1950-1990 in the 0-2000 m range than other studies, primarily due to the 700-2000 m depth interval, where data is sparser. Cooling of the deep ocean below 2000 m is only 26 ± 16 ZJ over 1950-1990, while our estimate of the 0-2000 m OHC change is about 27 ZJ less than the IAP estimate for this period, making the impact of the inclusion of the deep ocean below 2000 m roughly equal to the lower 0-2000 m OHC estimate we give versus the IAP. In the case of the JMA and NOAA estimates, it is clear that our lower estimate of shallow OHC change outweighs the impact of deep ocean cooling. That being said, our estimate of 0-2000 m OHC change from 1950-1990 is closer to the IAP estimate than the IAP estimate is to either the NOAA or JMA estimates, so it isn't so much that our estimate disagrees with the other estimates but that there is considerable disagreement across estimates in general.

In our revised manuscript, we have more clearly stated the precise impact of deep ocean cooling on the pre-1990 OHC trends, and we have also explicitly acknowledged the role of weaker upper-ocean warming during this time period as well. To clarify the impact of deep ocean cooling on the inferred OHC trends, we state how deep ocean cooling changes the inferred energy imbalance prior to 1990 determined by the ARANN (emphasis added): "Prior to 1990 the $dOHC/dt$ oscillates around zero, averaging -0.04 ± 0.11 $W m^{-2}$ for the period 1946-1990. *Without taking into account the deep ocean cooling during this period, the average $dOHC/dt$ would be 0.01 ± 0.09 $W m^{-2}$.*" We also discuss how including this deep ocean cooling into previous estimates would lower their inferred global ocean warming rates prior to 1990 (emphasis added): "Overall, the ARANN results support a broad consensus across almost all products of accelerated warming over time (Fig. 5), but they also suggest that previous estimates of ocean warming may have been biased too high prior to 1990, in part due to the neglect of deep ocean cooling. *Including the effects of deep ocean cooling, as determined by the ARANN, would lower rates of ocean warming prior to 1990 determined by previous objective mapping approaches^{11,12,13} by 18 to 32%.*"

We highlight the role of reduced upper-ocean warming in contributing to the ARANN's lack of warming in the discussion (emphasis added): "Nonetheless, deep ocean cooling does not entirely account for the near zero warming trend in OHC prior to 1990, especially when considering that the 0-700 m interval shows minimal change in the ARANN OHC estimate as well, averaging just 0.06 ± 0.08 $W m^{-2}$ from 1960-1990. *The difference between the ARANN and the IAP reconstruction¹¹ of OHC in the upper 2000 m, is similar in magnitude to the ARANN*

estimate of deep ocean cooling (Supplementary Fig. 15), and in general the spread across OHC estimates in the top 2000 m is larger than the deep ocean cooling trend estimated by the ARANN (Fig. 1b-c). This spread indicates large uncertainties related to methodological differences in estimating OHC over the latter half of the 20th century."

Furthermore, Supplementary Figure 14 indicates that shifts in ocean heat content by 50 ZJ can be caused by changes in the ocean climatology used by ARANN.

That is true, and in the revised manuscript we have explicitly called attention to this source of uncertainty (which is already included in our error bars). The revised manuscript now states "The large spread among OHC products prior to 1980 is primarily due to increased data sparsity in this period, but the choice of reference climatology also plays a role in enhancing uncertainties in the ARANN during this time period. ARANN OHC estimates for the 0-2000 m depth interval can vary by as much as 67 ZJ prior to 1970 when using different reference climatologies (Supplementary Figure 14), which is a source of uncertainty that has generally been neglected in the other mapping products."

These shifts are greater than the 30 ZJ decrease in deep ocean heat content that are central to main finding #2. Thus, these deep ocean temperature trends are not robustly indicated as the dominant cause of delayed acceleration.

The impact of adding the deep ocean to our 0-2000 m OHC estimate after controlling for climatology and instrumental bias correction is illustrated in the new Supplementary Fig 15. In general, the revised manuscript has better contextualized our findings by stating exactly how much the inferred deep ocean cooling contributed to the OHC trends prior to 1990 and after 1990, by highlighting the weaker warming trend in the upper ocean determined by the ARANN compared to previous estimates, and by highlighting the large uncertainties (partially due to the choice of climatology) in our estimated warming rates.

Summary remarks:

High-profile papers have the burden of being simultaneously novel, important, and statistically rigorous. Many such papers handle these constraints by doing minimal statistical analyses at the cost of weakening their impact. The authors are to be commended for completing a more quantitative statistical treatment; however, the presentation of the findings is not consistent with the quantitative results. I evaluate the deep ocean analysis to be interesting and valuable, but I cannot recommend publication in the current form.

We appreciate your commendation, as we have truly strived to produce a statistically rigorous analysis. As you point out, this comes with its own challenges. Taking into consideration your feedback, we have produced substantial revisions to our manuscript to ensure that our statements and interpretation do not exceed the confines of the analysis provided.

Additionally, we have clearly stated the major caveats and limitations of our findings. We have contextualized the contribution of the deep ocean cooling versus our lower upper-ocean OHC estimate to our finding of no OHC change prior to 1990. We have clarified our position that our finding of delayed ocean warming prior to 1990 is not statistically significant, given the findings of other studies and associated uncertainties across analyses. We have also clarified the impact that the choice of climatology has on the total warming estimate and the rate of warming in the early part of the record.

Minor issues:

Figure 2: the chosen baseline of the curves, which is ambiguous, will significantly affect the visual impression of the figure.

We have modified Fig. 1-2 and specify within the caption for Fig. 1 that “The zero anomaly is defined such that the mean OHC of the ARANN estimate for the period 1946-2019 is zero. The ARANN estimate is compared with the mean OHC anomaly from several other products that have been adjusted to the mean ARANN anomaly for 2005-2019”

We also state “The passive ocean heat uptake estimates are adjusted to the mean ARANN anomaly for 1955-1985. “

And “The RHS method gives a linear trend from 1985-2000 and from 2000-2015, which we have adjusted to the mean ARANN anomaly for 1985-2015.”

The Little Ice Age is most commonly defined to be a several hundred year interval.

This has been revised to note that the Little Ice Age covered the 14th-19th centuries.

Reviewer #2 (Remarks to the Author):

The paper ‘Closure of Ocean Heat Budget Reveals Delayed Acceleration of Earth’s Energy Imbalance’ has significantly improved after major revisions, and I recommend publication after further minor revision, see comments below.

Further Comments:

Title – and general comment as mention also below: The use of the wording ‘budget’ is confusing, and misleading. This study does not provide a budget analysis, it provides an evaluation of global ocean heat content. I strongly recommend to refine this wording as I suspect large confusion within the community when used in its current state. If the authors insist to keep the wording, a clear definition for what ‘budget’ stands for here, and why and how this study addresses a ‘budget closure’ analysis would need to be added for clarification.

To avoid this confusion, we have revised the title to "20th Century Cooling and Recent Warming of the Deep Ocean Contributed to Delayed Acceleration of Earth's Energy Imbalance" Please see the responses that follow. We have revised our terminology throughout to be more precise.

Other general comment (conclusion part): I strongly recommend to further discuss these new results in reflectance with previous estimates, and to particularly provide further links / ideas for needed research paths forward, and improvements and further evaluations needed.

Our revised discussion adds additional context for our findings within the broader scope of prior studies that may help guide future investigations.

For instance, we now state that “the spread across OHC estimates in the top 2000 m is larger than the deep ocean cooling trend estimated by the ARANN (Fig. 1b-c). This spread indicates large uncertainties related to methodological differences in estimating OHC over the latter half of the 20th century.” Whereas, there has been a “convergence of OHC estimates across methodologies” during recent decades, Therefore, a deeper investigation of the individual methodological choices made across studies should take precedent, since these potentially lead to large uncertainty across methods in the early record.

With the increasing contribution of the deep ocean to global OHC we also propose that “improved resolution of deep ocean temperature changes, will be key for developing accurate forecasts of Earth's energy budget and future climate change.”

P1, L11-12: There is a principal mis-concept in this statement. The EEI at TOA is currently measured by remote sensing. There is a challenge to obtain the absolute value with the CERES program (anchored by ocean heat content), but it is the most precise measure today on the variations of the EEI. Then, the heat inventory to obtain the heat stored in the Earth system from a positive EEI is derived from the Earth heat inventory, and as stated, the majority is stored in the ocean. Revision is needed accordingly.

We have revised this statement to focus on the historical evolution of EEI, which cannot be measured by remote sensing: “The historical evolution of Earth's energy imbalance can be quantified by changes in the global ocean heat content. However, historical reconstructions of ocean heat content often neglect a large volume of the deep ocean, due to sparse observations of ocean temperatures below 2000 m.”

P1, L14-15: Which is the percentage today estimated from the deep ocean contribution? More than the 10% which are missed out by considering the ocean only in this concept for the Earth heat inventory, particularly when also accounting for the uncertainties?

We find that the deep ocean below 2000 m warmed by 48 ± 19 ZJ from 1990-2019, which represents between 7 to 21% of the total ocean warming over this period at 2 sigma confidence intervals. von Schuckmann et al. (2020) provides an updated analysis of contributions to the Earth heat inventory. They estimate roughly 7 ± 1 ZJ of warming for the cryosphere, 16 ± 8 ZJ for land, and 3.6 ± 0.5 ZJ to the atmosphere for a total of $\sim 27 \pm 8$ ZJ for 1990 onwards. By our

estimate, the deep ocean then accounts for between 7 to 20% of the total Earth heat inventory since 1990 and non-ocean components account for 6-11%.

The revised manuscript puts the heat gain in the deep ocean in context of that in the upper ocean: " Additionally, the ARANN results suggest that the deep ocean below 2000 m has added 48 ± 19 ZJ since 1990, or about 10 to 28% of the ocean warming above 2000 m during this period, significantly contributing to the accelerating EEI in recent decades. This contribution is likely larger than that from non-ocean components of the Earth energy budget, including the land surface, cryosphere, and atmosphere, which together account for $\sim 27\pm 8$ ZJ of warming since 1990".

P2, L45-48: This statement is not linked to any reference, and also it is not clear on how the energy budget closure is approached here: Is it the close correspondence between net flux at TOA and the Earth heat inventory? If yes, I am puzzled because I do not see recent publications pointing to a new 'missing energy concept' (DOI: 10.1126/science.1187272) at timescales the authors are aiming to resolve (long-term) – where is this statement – and the major motivation of the study (incl. the title) based on? If the EEI budget constraint is not addressed here, the concept on 'energy budget' in this study needs to be clarified.

By "open ocean heat budget" we were referring to the fact that the heat budget of the ocean was incomplete, because the heat content below 2000 m had not been completely mapped out. However, since that wording caused confusion, we have eliminated it. This does not affect the framing of the results, only the words we use to describe them.

P2, L47-48: this does not leave the energy budget open, but deep ocean contributions to global ocean energy storage remains uncertain – this would be a correct way to phrase this, see comment above.

We have revised our wording to say "the deep ocean below 2000 m remains poorly observed, even during the Argo era, which leads to additional uncertainty on current estimates of total warming."

P4, L133: Would recommend to add the uncertainty range to this estimate of 30-40 ZJ – general recommendation for the entire document, will not raise every time.

Error bars have been added to estimates throughout the revised text.

P7, L295 recommend to use wording 'quasi-equilibrium' instead

We have revised this accordingly

P8, L320-321: recommend to add also the recent paper of Kramer et al. into this discussion (<https://doi.org/10.1029/2020GL091585>)

Observations of recent increasing radiative forcing provide additional valuable context here. We have incorporated this study into our revised discussion.

Reviewer #3 (Remarks to the Author):

I appreciate the author's further sensitivity analysis to the choice of climatology, and incorporation of different climatologies into the OHC estimations presented in the study. I also thought the revised passages of the manuscript were much clearer and easier to place in the context of previous estimates. I am generally satisfied with the revision and still feel the paper is appropriate for the Nature readership. I have one remaining issue with the paper, and that is, despite substantial analysis of error, there is relatively little discussion or disclosure of it in the text. Simply adding error ranges into estimates, and a few disclaimers, would strengthen the reader's understanding of where the methods are more or less certain.

Thank you for this feedback. We have expanded our disclaimers in relation to sources of uncertainty. We have also added error bars where appropriate throughout the text to clarify the level of uncertainty. See also our responses to reviewer #1 and #2.

Prior to the results we now also clarify our reported uncertainties with "Four instrumental bias corrections and six decadal climatologies are combined with random selections of temperature data to produce the 240 ensemble members used in this study. This ensemble is used to assess the uncertainty of our OHC reconstruction and provide bounds on our estimates of ocean warming. All estimated warming rates come from fitting a linear trend to the mean ARANN OHC estimate and uncertainties in these rates are calculated by taking 2σ across all ensemble members."

To help with interpretability of the uncertainty in presented figures, we also state "for simplicity, warming estimates from other studies are not plotted with their respective confidence intervals, as the methodology for calculating these varies by study, but they generally possess uncertainty levels similar to those provided by the ARANN."

Additionally, we add points in the text that speak to additional uncertainties across methods and contextualize our uncertainties, such as "Prior to 1980, the mapping methods diverge somewhat in their predictions over the upper 2000 m. The ARANN reconstruction on average estimates approximately 22-37 ZJ less warming above 2000 m than the other objective mapping products from 1955-1980. The disagreement is most pronounced for the earliest time periods. Adjusting the objective mapping estimates to the mean ARANN 0-2000 m OHC value over 2005-2019, the ARANN OHC in 1955 is 71 ± 58 ZJ higher than that estimated by the IAP¹¹ product, 51 ± 58 ZJ higher than the NOAA¹² product, and 114 ± 58 ZJ higher than the JMA¹³ product (Fig. 1b-c). The large spread among OHC products prior to 1980 is primarily due to increased data sparsity in this period, but the choice of reference climatology also plays a role in enhancing uncertainties in the ARANN during this time period. Mean ARANN OHC estimates for the 0-2000 m depth interval can vary by as much as 67 ZJ prior to 1970 when using different reference

climatologies (Supplementary Figure 14), which is a source of uncertainty that has generally been neglected in the other mapping products. “

Specific points:

L99, L101, L133, L144 added error estimates (+/-).

We have added the appropriate error estimates.

L301: “so the lack of global ocean warming...” add something along the lines of “implied by our new methodology.”

The statement in full now reads “Anthropogenic radiative forcing has remained positive and continued to grow in magnitude over the past century¹, so the lack of global ocean warming implied by the ARANN results over the period from 1950-1990 may seem counterintuitive at first.”

L333: I’m not sure I’d say there’s relatively little impact- earlier in the record, the average implied OHC anomaly is ~50ZJ different between climatologies, basically the same as the total OHC anomaly at during that time in Fig. 1. Please amend to something more clear- the climatology has a small impact over the final 3 decades and a more significant impact, contributing to the greater uncertainty in the OHC estimate, earlier in the record and deeper in the ocean.

Our revised statement reads "The choice of climatology produces a difference in the total OHC change from 1946-2019 of 49 ZJ. The impact of climatological choice is small for the final three decades of the OHC record but has a more significant impact on the global OHC estimate in the deep ocean and further back in time (see Supplementary Fig. 14)."

REVIEWERS' COMMENTS

Reviewer #1 (Remarks to the Author):

The authors, Bagnell and DeVries, have addressed my review and placed the manuscript in an improved context relative to other works. Three minor issues are listed below.

L46: MAKES it

Paragraph beginning at L120: Consider clarifying this description because only differences in ocean heat content are thermodynamically interpretable in this context.

L834-835: are the color labels for OPT-0015 and GF reversed?

Reviewer #1 (Remarks to the Author):

The authors, Bagnell and DeVries, have addressed my review and placed the manuscript in an improved context relative to other works. Three minor issues are listed below.

L46: MAKES it

Revised

Paragraph beginning at L120: Consider clarifying this description because only differences in ocean heat content are thermodynamically interpretable in this context.

We have modified our description in this paragraph to make it more apparent that we are considering changes/ differences in OHC from one time to another.

L834-835: are the color labels for OPT-0015 and GF reversed?

Yes, this was an oversight that has been remedied in the revised manuscript.